# Loss of PGC-1α in RPE induces mesenchymal transition and promotes retinal degeneration

Mariana Aparecida Brunini Rosales[1,2] , Daisy Y Shu[1,2] , Jared Iacovelli[1,2], Magali Saint-Geniez[1,2]

The retinal pigment epithelium (RPE) supports visual processing and photoreceptor homeostasis via energetically demanding cellular functions. Here, we describe the consequences of repressing peroxisome proliferator-activated receptor γ coactivator-1 α (PGC-1α), a master regulator of mitochondrial function and biogenesis, on RPE epithelial integrity. The sustained silencing of PGC-1α in differentiating human RPE cells affected mitochondria/autophagy function, redox state, and impaired energy sensor activity ultimately inducing epithelial to mesenchymal transition (EMT). Adult conditional knockout of PGC-1 coactivators in mice resulted in rapid RPE dysfunction and transdifferentiation associated with severe photoreceptor degeneration. RPE anomalies were characteristic of autophagic defect and mesenchymal transition comparable with the ones observed in age-related macular degeneration. These findings demonstrate that PGC-1α is required to maintain the functional and phenotypic status of RPE by supporting the cells' oxidative metabolism and autophagy-mediated repression of EMT.

## Introduction

The retina pigment epithelium (RPE) is a highly specialized monolayer of pigmented cuboidal cells separating the neural retina from the choroidal vasculature and serving critical roles to maintain retinal homeostasis and visual function. Integrity of the RPE epithelial phenotype is crucial for the cells to execute their functions, which include visual pigment recycling, photoreceptor outer segment (POS) phagocytosis, secretion of neurotrophic factors and cytokines, light absorption, ion and water transport, and maintenance of the outer blood–retinal barrier (Strauss, 2005).

Like most postmitotic cells, RPE functional maturation depends on the induction of a specific metabolic program able to support their energetic requirements. In vitro maturation of human RPE cells was found to be correlated with increased mitochondrial biogenesis and oxidative metabolism (Adijanto & Philp, 2014; Iacovelli et al, 2016). Establishment of this metabolic program

appears to be, at least in part, controlled by peroxisome proliferator–activated receptor γ coactivator 1α (PGC-1α), a master regulator of mitochondrial biogenesis and function (Wu et al, 1999; Fernandez-Marcos & Auwerx, 2011), whose expression in RPE increases in alignment with their metabolic and functional maturation (Iacovelli et al, 2016). Moreover, PGC-1α gain-of-function was shown to enhance RPE metabolic functions and resistance to oxidative stress through the induction of oxidative metabolic genes and antioxidant enzymes (Iacovelli et al, 2016). Additional evidence that oxidative metabolism is essential for the maintenance of RPE functional and morphological integrity comes from recent clinicopathologic analysis of age-related macular degeneration (AMD). AMD, in particular its dry form, is characterized by focal RPE dysfunction and/or atrophy leading to the degeneration of the overlying photoreceptor and focal vision loss. Decreased expression or signaling of key components of the RPE antioxidant mechanism and increase in mitochondrial DNA damage from oxidative stress have been described in aged and AMD eyes (Lin et al, 2011; Golestaneh et al, 2016). Furthermore, mitochondrial defects and increased sensitivity of oxidative damage associated with PGC-1α repression were also observed in iPS-derived RPE cells from dry AMD patients (Golestaneh et al, 2016), suggesting that RPE metabolic alteration and oxidative damage act as key initiating events in the activation of the various degenerative mechanisms associated with AMD pathogenesis. Because PGC-1α is a major regulator of mitochondrial function, it may also be an important player in AMD pathogenesis (Kaarniranta et al, 2018) and global knockout mouse models for PGC-1α have been found to mimic some of the degenerative processes characteristic of human AMD (Egger et al, 2012; Zhang et al, 2018). However, a recent study using cultures of primary RPE cells from AMD donors provided conflicting results as diseased cells were found to express higher levels of PGC-1α and to be more resistant to oxidative stress (Ferrington et al, 2017). Although a direct implication for PGC-1α in AMD pathogenesis remains to be determined, it is clear that its expression and function are tightly correlated with RPE viability and function.

The interplay between cellular metabolism and autophagy is emerging as a critical regulatory mechanism for maintaining epithelial cell integrity and actively repressing epithelial to mesenchymal transition (EMT) (Gugnoni et al, 2016; Morandi et al, 2017).

---

[1]Schepens Eye Research Institute of Massachusetts Eye and Ear, Boston, MA    [2]Department of Ophthalmology, Harvard Medical School, Boston, MA

Correspondence: magali_saintgeniez@meei.harvard.edu

EMT of RPE cells is central to numerous degenerative ocular diseases, including proliferative vitreoretinopathy (PVR) (Machemer et al, 1978; Casaroli–Marano et al, 1999) and AMD (Hirasawa et al, 2011). It is well established that mitochondrial dysfunction can drive epithelial cell degeneration and EMT through down-regulation of oxidative metabolism, increase of ROS production, and mtDNA damage (Guerra et al, 2017). Autophagy, which is controlled by the AMPK/mTOR energy sensor signaling axis, can also directly regulate EMT molecular reprogramming by selective degradation of major EMT molecular switches (Qiang et al, 2014; Bertrand et al, 2015; Grassi et al, 2015).

As a key regulator of oxidative metabolism and mitochondrial health, PGC-1α is likely involved in controlling RPE autophagic function as shown in skeletal muscle cells (Vainshtein et al, 2015). Indeed, PGC-1α was found to be transcriptionally induced to support POS phagocytosis (Roggia & Ueta, 2015). To further determine the critical contributions of PGC-1α in coordinating both energy metabolism and epithelial phenotypic features of RPE, we examined the in vitro and in vivo consequences of PGC-1α deletion on RPE metabolic, autophagic, and epithelial status.

# Results

## PGC-1α repression causes RPE mitochondrial dysfunction and oxidative stress

To investigate the role of PGC-1α in RPE metabolism, integrity, and function, we silenced PGC-1α in the human RPE cell line ARPE-19 by lentivirus-mediated delivery of shRNA. Efficient repression of PGC-1α expression in GFP+ ARPE-19 cells (Fig S1A) was confirmed at the mRNA (≈96% loss, Fig 1A) and protein levels (≈78% loss, Fig 1B) after in vitro maturation for 7 d, the time of maximal PGC-1α induction in differentiating parental cells (Iacovelli et al, 2016). First, we evaluated the effect of silencing PGC-1α on mitochondrial morphology and metabolic function in shPGC-1α ARPE-19 cells. MitoTracker staining in cells matured for 7, 14, and 21 d showed severe disorganization of the mitochondrial network associated with PGC-1α silencing characterized by loss of tubular organization and formation of donut/blob-shaped mitochondria, a hallmark of mitochondrial dysfunction (Ahmad et al, 2013) (Fig 1C). Such transition from tubular to globular shape was correlated with the transcriptional repression of the mitochondrial fission/fusion dynamics genes FIS1 and MFN2 (Fig 1D). Change in mitochondrial mass was evaluated by quantifying the activity of citrate synthase, the enzyme catalyzing the first reaction of the Krebs cycle, and a superior biomarker of mitochondrial content (Steen et al, 2012). Despite the well-established function of PGC-1α in promoting mitochondrial biogenesis, citrate synthase activity (CSA) was found to be significantly higher in shPGC1A cells at all time-points studied (Fig 1E), whereas the expression of the mitochondrial replication-related genes POLG and TFAM remained unchanged except for a small induction of TFAM after 21 d (Fig S1B). This unexpected increase in CSA activity is likely reflecting an adaptive response to metabolic/oxidative stress as previously shown in pancreatic cancer (Schlichtholz et al, 2005; Vishnyakova et al, 2016). Indeed, evaluation of mitochondrial OXPHOS by extracellular flux analysis

and measurement of the oxygen consumption rate (OCR) revealed that loss of PGC-1α decreased all phases of respiration as early as day 7 (Fig 1F). Conversely, maximal and reserved glycolytic capacities of shPGC-1α cells were significantly increased (Fig S1C and D). Although shPGC-1α cells display enhanced glycolytic capacity, cellular ATP content was lower at all time points (Fig S1E). These changes were accompanied by increased mitochondrial superoxide production after day 14 (Fig 1G), whereas general ROS content was higher at all time points (Fig 1H). ROS accumulation in PGC-1α–silenced cells was associated with repression of catalase (CAT) expression at days 7, 14, and 21 (Fig 1I), whereas cytoplasmic superoxide dismutase (SOD1), mitochondrial thioredoxin (TXN2), and glutathione peroxidase (GPX) genes were found to be significantly down-regulated at day 21 (Fig 1I). Repression of antioxidant enzymes was also associated with down-regulation of the mitochondrial sirtuin (sirtuin 3, SIRT3), a known target of PGC-1α transcriptional induction with important roles in ROS suppression and mitochondrial function (Kong et al, 2010) (Fig 1J). Surprisingly, sirtuin-1, a key upstream regulator of PGC-1α activity (Cantó & Auwerx, 2009), was also found to be repressed after sustained PGC-1α loss of function (Fig 1J). To determine if some of the changes observed could be attributed to potential compensatory induction of PGC-1 isoforms, gene expression of PGC-1β and PRC were quantified and found to not be statistically different between shPGC1A and shCtrl cells (Fig 1K). Together, these findings indicate that PGC-1α is a critical regulator of RPE mitochondrial dynamics, function, and cellular redox state.

## PGC-1α silencing impairs RPE autophagic flux

RPE homeostasis and functions are tightly dependent on regulated autophagic activity (Kaarniranta et al, 2013) and failure to degrade damaged cellular organelles and molecules under pro-oxidative conditions can trigger RPE dysfunction (Saadat et al, 2014; Zhang et al, 2015; Golestaneh et al, 2017). To evaluate the effect of PGC-1α silencing on the lysosomal/autophagic status of RPE cells, we first examined the organization of late-endosomes stained with lysosomal-associated membrane protein 1 (LAMP-1) during RPE maturation and observed at all time points the presence of highly dilated LAMP-1+ lysosomes in shPGC1A cells compared with control cells, where endosomes maintained a uniformly small size (Fig 2A). Based on these observations, we examined whether expression of autophagy-associated genes could be affected in shPGC1A cells and found that MAP1LC3B and WIPI, both involved in the initiation and lengthening of the autophagosomes, were suppressed from day 7, whereas ATG4D, involved in late stages of autophagosome maturation and ATG9B, which modulates the lysosomal degradation pathway, was reduced at 14 and 21 d compared with control conditions (Fig 2B). Contrariwise, adenovirus-mediated PGC-1α overexpression strongly induced the expression of autophagy-related genes in parental ARPE-19 (Fig 2C). To provide further evidence of a direct role for PGC-1α in regulating RPE autophagic activity, parental ARPE-19 cells were subjected to amino acid starvation (AAS) for 24 h to promote autophagic flux (Martina et al, 2016) and PGC-1α was found to be significantly induced along with other genes implicated in autophagy and lysosomal biogenesis (Fig 2D). We next evaluated the consequences of PGC-1α

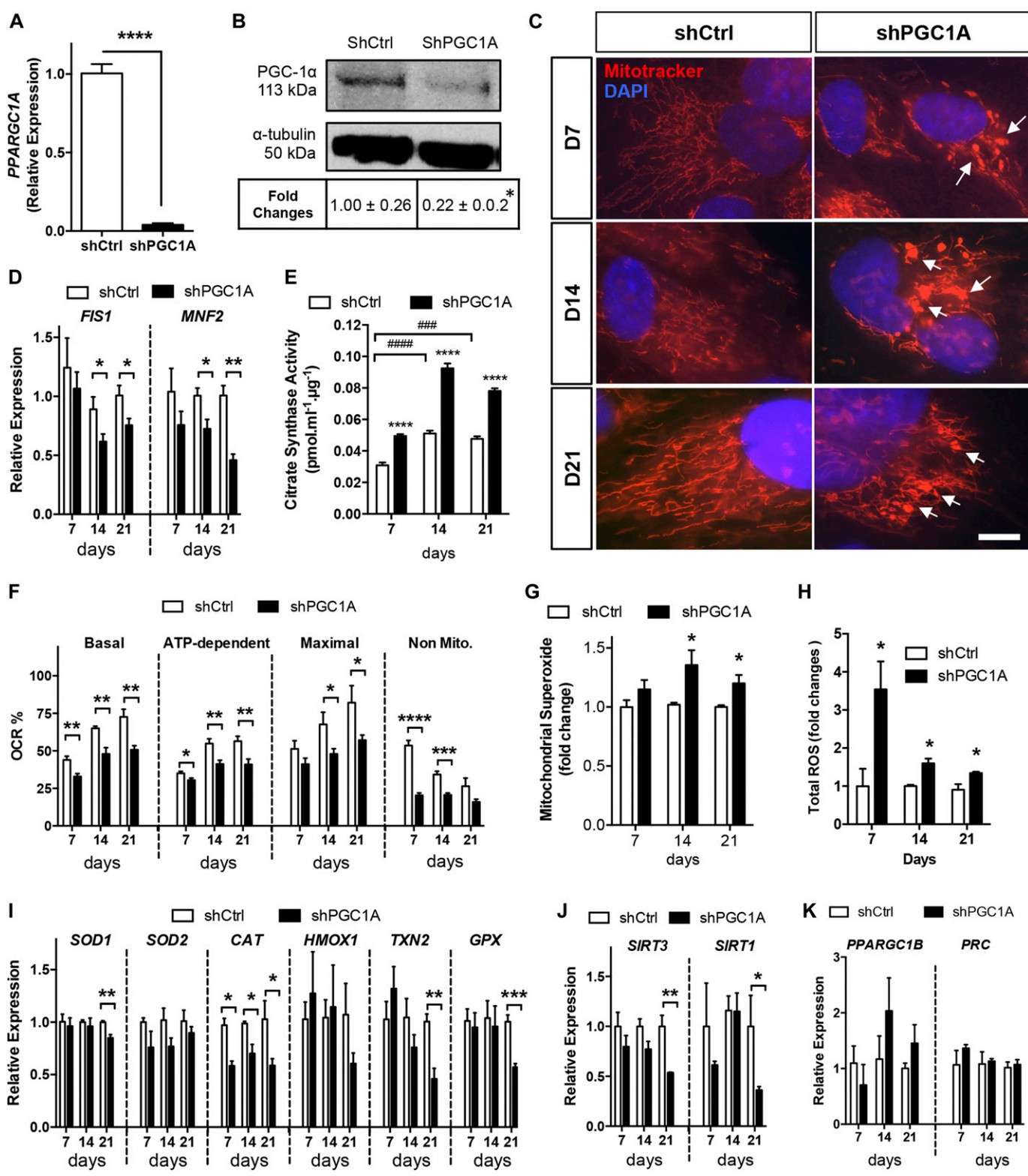

**Figure 1. PGC-1α silencing in RPE cells promotes mitochondrial dysfunction and oxidative stress.**
**(A, B)** Validation of PGC-1α silencing at the mRNA and protein levels by quantitative RT-PCR (A) and Western blotting (B) in ARPE-19 cells matured for 7 d (n = 3–4 per condition, respectively). **(C)** Evaluation of mitochondrial morphology using MitoTracker (red) and DAPI (nuclei, blue) co-staining showing prominent shape changes from tubular to "donut-blob" formation in shPGC1A cells (arrows). Scale bar is 10 $\mu$m. **(D)** Quantification of the mitochondrial dynamics genes FIS1 and MFN2 by qPCR (n = 3–4). **(E)** CSA quantification (n = 4). **(F)** OCR parameter measurements of shPGC1A and shCtrl ARPE-19 at day 7, 14, and 21 (n = 5). **(G, H)** Quantification of mitochondrial superoxide production using MitoSox Red (n = 5–7) (G) and cytoplasmic ROS content with DCFH-DA (n = 3–6) (H). **(I)** Relative expression of antioxidant enzyme

loss-of-function on RPE autophagic flux, by monitoring the conversion of LC3 from the soluble cytoplasmic form LC3-I to the lipid-bound form LC3-II (Jiang et al, 2015) after 2 h of AAS, the optimal time point for LC3-I to LC3-II conversion in parental ARPE-19 cells (Fig S2A). Whereas the expected increase in LC3-II/LC3-I ratio was observed in starved shCtrl cells, no LC-3 conversion was detected in shPGC1A cells (Fig 2E), suggesting that in the absence of PGC-1α, the cells were no longer able to induce autophagy to maintain an homeostatic amino acid content. Assessment of the autophagic flux by use of the lysosomal inhibitor chloroquine (CQ) revealed that, contrary to shCtrl cells, LC3-II level in shPGC1A cells was not increased by CQ (Fig S2B), confirming that silencing PGC-1α impairs RPE autophagic flux. The defective autophagic response in shPGC1A cells was further demonstrated by accumulation under both basal and AAS conditions of the ubiquitin-binding adaptor proteins such as p62/SQSTM1 that binds LC3 and is, therefore, degraded by autophagy (Fig 2F). Activation of AMPK, the major regulator of autophagy initiation, measured as the ratio of phospho-Th172 to total AMPKα, was increased in shCtrl cells exposed to AAS but not in shPGC1A cells (Fig 2G), showing that PGC-1α is critical for amino acid–dependent AMPK activation. As susceptibility to oxidative stress and impaired autophagy was observed to be accompanied by the up-regulation of AMD-associated genes such as apolipoprotein E (APOE) expression in primary RPE cells from AMD patients (Golestaneh et al, 2017), we quantified the expression of APOE and found its protein level increased in shPGC-1α RPE cells at 14 d (Fig 2H).

### Sustained PGC-1α silencing induces RPE dedifferentiation and mesenchymal transition

To investigate the phenotypic consequences of the mitochondrial and autophagic dysfunction triggered by PGC-1α silencing, integrity of the RPE barrier was assessed by immunostaining of the tight junction protein ZO-1, normally localized at the cell membrane of mature RPE. When compared with shCtrl cells, ZO-1 was found to be disorganized and limited to no membranous localization at days 14 and 21 in shPGC1A (Fig 3A). Failure of shPGC1A RPE cells to form and maintain proper tight junctions was confirmed by measuring transepithelial electrical resistance (TER) of the cells (Fig 3B). As progressive loss of barrier function may be caused by epithelial dedifferentiation, RPE monolayers were stained for vimentin whose induction is associated with RPE mesenchymal transition (Adijanto et al, 2012) and oxidative/nitrosative stress (Sripathi et al, 2012). Whereas the vimentin intermediate filaments (IFs) formed an evenly distributed arborized network in shCtrl cells, gradual disorganization and condensation with prominent perinuclear bundling of the IFs was observed in shPGC1A cells (Fig 3C). We then asked whether the observed collapse and entanglement of vimentin IFs could be indicative of EMT. Gene expression analysis showed induction of vimentin expression at days 14 and 21 and of the EMT

transcription factors *ZEB1* and *TWIST1* at day 21 concomitant with repression of *P53* (Fig 3D). Protein expression analysis in cells matured for 21 d confirmed the significant induction of ZEB1 and Twist (Fig 3E and F) along with mTOR-dependent pS6 ribosomal protein (Ser93) activation (Fig 3G) and decreased total levels of AMPK-α (Fig 3H) in shPGC1A cells. Importantly, this phenomenon was associated with unregulated cell division as indicated by increased cell density in PGC-1α–silenced cells compared with nonsilenced controls (Fig 3I). Moreover, assessment of cellular proliferation by quantification of proliferating cell nuclear antigen (PCNA) levels indicated higher proliferation in shPGC-1α cells at day 7 but not days 14 and 21 (Fig S2C). Taken together, these data indicate that PGC-1α is required to maintain RPE epithelial phenotype and that sustained PGC-1α loss triggers EMT in RPE.

### PGC-1 deletion in adult mice causes RPE dysfunction and neurodegeneration

To validate our findings in vivo, we developed and characterized a mouse model of conditional PGC-1α deletion in adult RPE by AAV-Cre recombination of PGC-1–floxed animals. To bypass any potential compensatory responses from PGC-1β (Szczelecki et al, 2014), which is normally expressed at low to undetectable levels and inversely correlated to PGC-1α levels in matured RPE (Iacovelli et al, 2016), we elected to ablate both isoforms using double PGC-1α/β–floxed animals (Rowe et al, 2013). The RPE specificity of our transgene expression was achieved by subretinal delivery and titration of the RPE trophic AAV2/8 (Vandenberghe et al, 2011). At 2- and 4-wk post-AAV injection in experimental floxed animals, we used fundus photography (Fig S3A) and RPE/choroid flat mounts to confirm the efficient transduction by AAV2/8-CASI-GFP (AAV-GFP) and AAV2/8-CASI-Cre (AAV-Cre) of most of the RPE area (Fig 4A). Costaining for GFP, F-actin, and Cre on RPE flat mounts showed that 1-mo after AAV-GFP or AAV-Cre injection, the RPE monolayer maintained a normal cuboidal morphology and that Cre expression was only detected in the nuclei of RPE from AAV-Cre–injected animals (Figs 4A and S3B). Gene expression analysis performed on isolated RPE and neuroretinal tissues showed reduction by 80% of PGC-1α mRNA in the RPE of AAV-Cre–injected eyes compared with AAV-GFP controls, whereas PGC-1β was undetectable in both conditions and unchanged in the retina (Fig 4B). Immunodetection of GFP+ cells on ocular cryosections from AAV-GFP–injected mice confirmed the RPE trophism of the virus under our conditions (Fig 4C).

Monitoring of retinal morphology by optical coherence tomography (OCT) and retinal thickness measurement did not reveal any significant changes in AAV-Cre mice 1 mo postinjection besides some loss of retinal lamination (Fig 5A and B), but scotopic electroretinogram (ERG) recording showed a trend, although non-statistically significant, toward a reduction of the *a*- and *b*-wave amplitudes, whereas the RPE generated *c*-wave was decreased by ≈66% (Fig S3C). At 4 mo, a significant decrease in total retinal

---

genes—catalase (CAT), cytosolic superoxide dismutase (SOD1), mitochondrial thioredoxin (TXN2), and glutathione peroxidase (GPX)—analyzed by qPCR (n = 3–4). **(J)** Sirtuins (SIRT) enzymes 1 and 3 gene expression analysis by qPCR (n = 3–4). **(K)** PGC-1 isoform gene expression, PGC-1β, and PRC (n = 3). Error bars are means ± SEM. Data were analyzed by multiple unpaired *t* test comparisons using the Holm–Sidak method (A–D, F–K) or ANOVA followed by Tukey's multiple comparison test (E). *$P \le 0.05$; **$P \le 0.01$; ***$P \le 0.001$; ****$P \le 0.0001$ compared with their respective shCtrl at each time points. ###$P \le 0.001$; ####$P \le 0.0001$ compared with day 7 shCtrl.

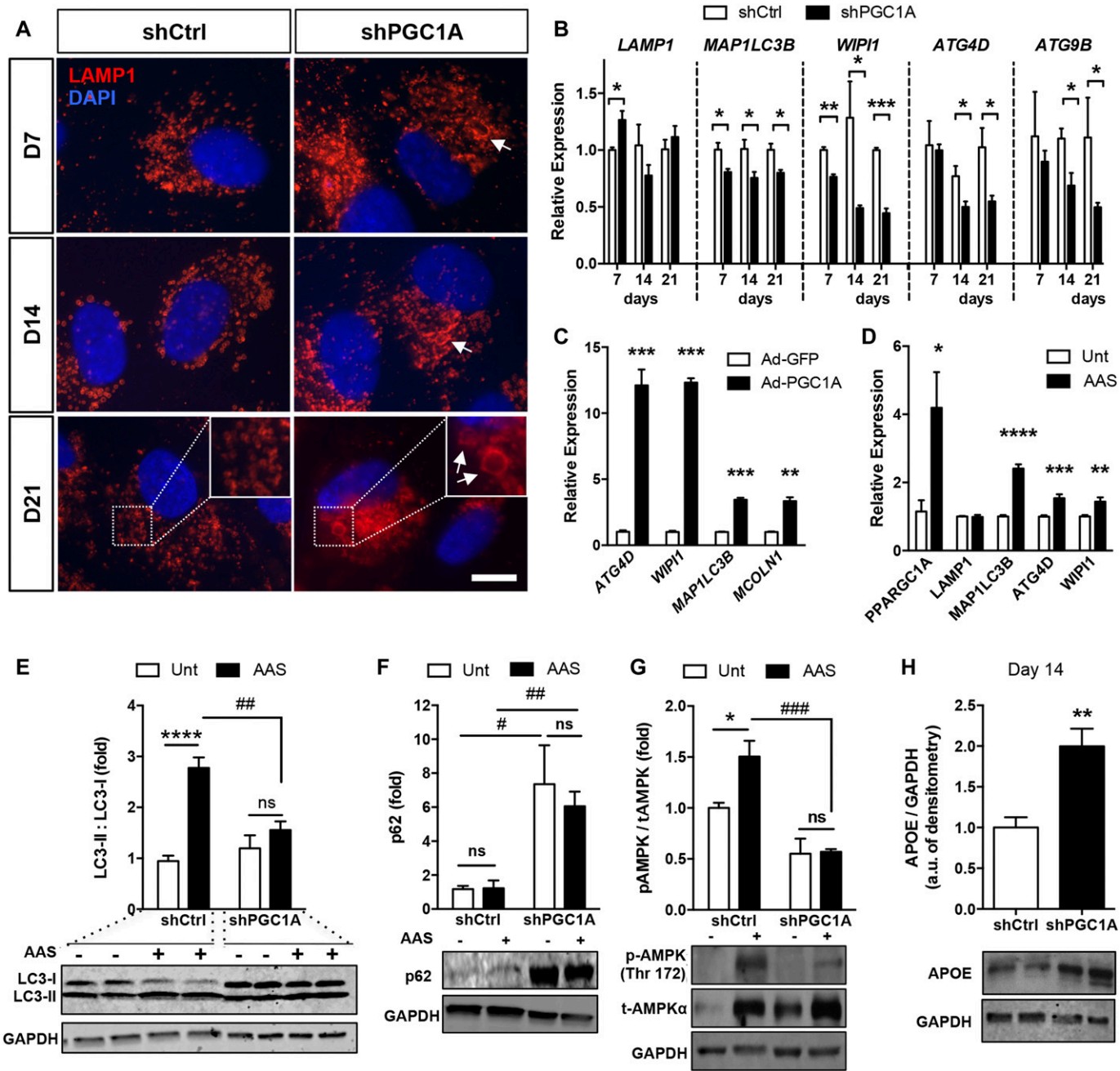

**Figure 2. PGC-1α activation is required for autophagy flux initiation in ARPE-19 cells.**
**(A)** Immunodetection of late-stage endosomes (LAMP-1, red) in shPGC1A and shCtrl ARPE-19 matured for up to 21 d. Arrows indicate dilated late endosomes in shPGC1A cells. **(B)** Expression profiles of autophagy-related genes MAP1LC3B and WIPI (initiation of autophagosomes), ATG4D (autophagosome maturation), and ATG9B (lysosomal degradation pathway) (n = 3–4, per condition and time points). Scale bar is 10 μm. **(C)** Changes in autophagosome biogenesis-related genes ATG4D, WIPI1, MAP1LC3B, and MCOLN1 in parental ARPE-19 cells 24 h after adenoviral-mediated PGC-1α overexpression (n = 3). **(D)** PGC-1α and autophagy-related genes LAMP1, MAP1LC3B, ATG4D, and WIPI1 expression quantified by qPCR in parental ARPE-19 cells after AAS for 24 h. Untreated (Unt) cells were used as controls (n = 4). **(E–G)** Western blot analysis for LC3B (E); polyubiquitin-binding p62 protein sequestosome (p62) (F); phospho (Thr172) and total AMPKα (G) in shCtrl and shPGC1A cells 2 h after AAS. Immunoblots are representative of n = 3–4 experiments per conditions. **(H)** APOE protein expression in shCtrl and shPGC1A cells ARPE-19 cells after 14 d of maturation (n = 4). For all blots, GAPDH was used as the loading control. Error bars are means ± SEM. Data were analyzed by multiple unpaired t test comparisons using the Holm-Sidak method (B–D, H) or ANOVA followed by Tukey's multiple comparison test (E–G). *P ≤ 0.05; **P ≤ 0.01; ***P ≤ 0.001; ****P ≤ 0.0001 versus respective controls. #P ≤ 0.05, ##P ≤ 0.01 and ###P ≤ 0.001 = shPGC1A versus shCtrl under AAS.

thickness measured by OCT was observed in AAV-Cre–injected PGC-1–floxed mice compared with AAV-GFP group (Fig 5A and B). Functional evaluation of the retina and RPE by ERG revealed severe

reduction of the a-, b-, and c-wave amplitudes and delayed peak time of response (Fig 5C–F). As a control, OCT, and ERG analysis were also conducted in wild-type C57BL/6J injected with AAV-GFP and

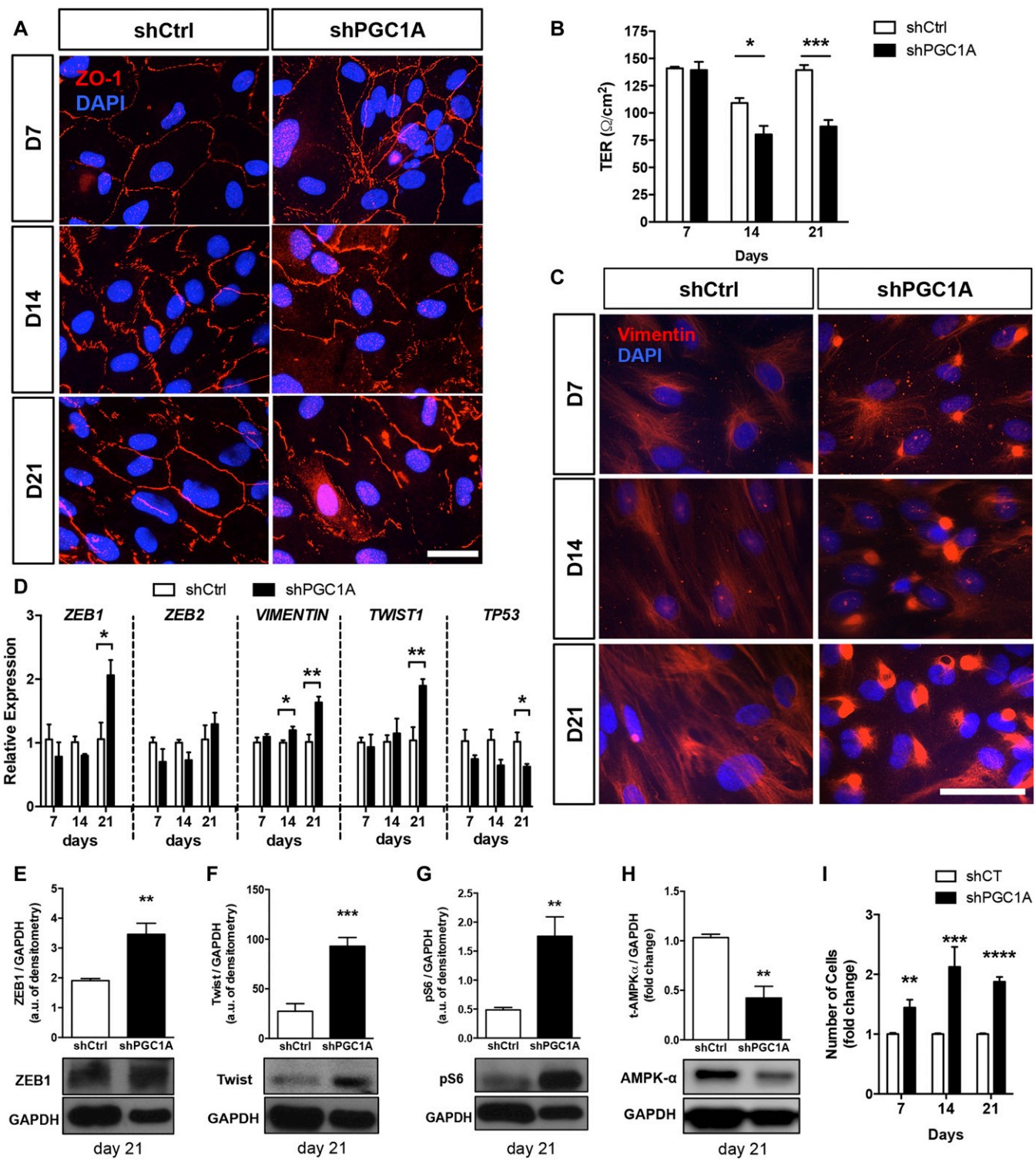

**Figure 3. Sustained PGC-1α loss of function triggers mesenchymal transition.**
**(A)** Immunodetection of the tight junction protein ZO-1 (red) in shPGC1A and shCtrl ARPE-19 cells matured for up to 21 d. Scale bar is 10 μm. **(B)** Barrier integrity measured by TER (n = 3 per condition). **(C)** Immunostaining of the vimentin IFs (red) in shPGC1A and shCtrl ARPE-19 cells matured from 7 up to 21 d. DAPI (blue) was used to stain nuclei. Scale bar is 50 μm. **(D)** Expression of EMT-related genes ZEB1, ZEB2, VIM, TWIST, and the cell cycle regulator p53 measured by qPCR analysis (n = 3–4). **(E–H)** Protein levels of the EMT transcriptional factors ZEB1 (E) and TWIST1 (F), the mTOR-dependent phospho-S6 ribosomal protein (Ser235/236) (G) and total AMPKα (H) in shPGC1A and shCtrl cells at 21 d. Immunoblots are representative of n = 3–4 samples per conditions. **(I)** Quantification of cell number per well expressed as fold change to shCtrl for each time point demonstrating significant increases in cell density in shPGC1A cells (n = 8–15). Error bars are means ± SEM. Data were analyzed by unpaired $t$ test. *$P \leq 0.05$; **$P \leq 0.01$; ***$P \leq 0.001$; ****$P \leq 0.0001$ compared with their respective shCtrl at each time points.

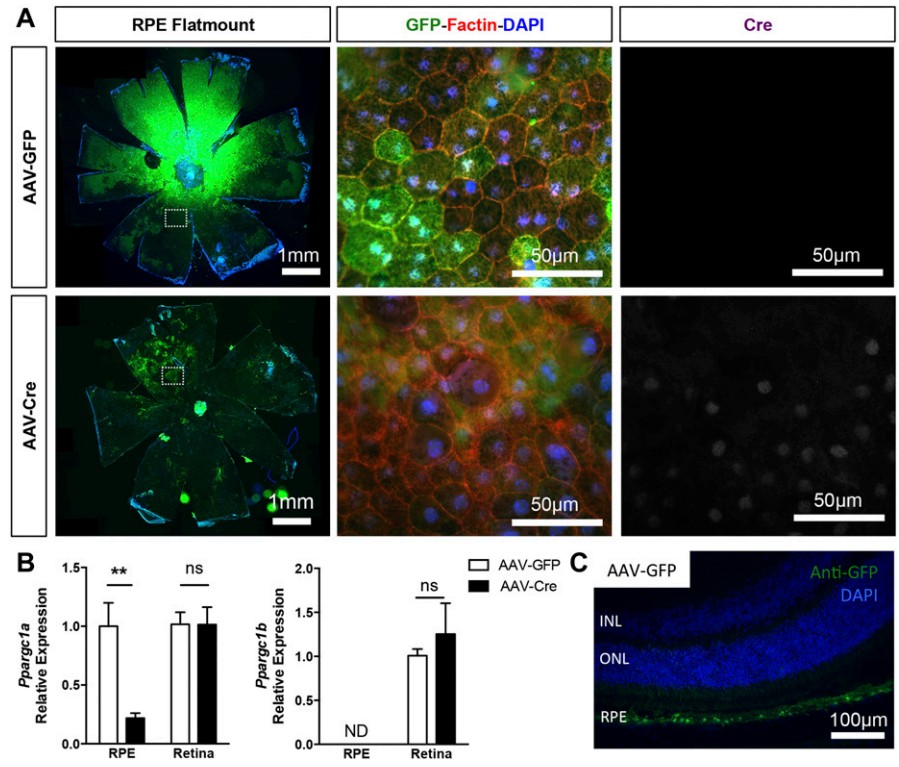

**Figure 4. RPE-specific deletion of PGC-1 isoforms in adult mice.**
**(A)** RPE/choroid flat-mounted preparations from adult PGC-1α;PGC-1β–double floxed mice 1 mo after subretinal injection with AAV-GFP and AAV-Cre. Immunolocalization of the GFP reporter (green) confirmed efficient viral transduction of the entire RPE surface. High magnifications in the mid-periphery (dotted squares) show GFP+ RPE cells labeled with F-actin (red). Cre expression (white) was only detected in the nuclei of RPE from AAV-Cre–injected animals. **(B)** PGC-1α and PGC-1β gene expression analysis in RPE/choroid and retinal tissues from AAV-GFP and AAV-Cre mice 1 mo postinjection (n = 3–6). **(C)** Representative ocular cryosection from experimental mice 4 mo after AAV-GFP injection, showing highly selective viral transduction of the RPE cells. Scale bar is 100 μm. Error bars are means ± SEM. Data were analyzed by unpaired *t* test. **P ≤ 0.01 compared with their respective AAV-GFP control mice.

AAV-Cre and showed no alteration in morphology (Fig S4A–C) and function (Fig S4D).

## PGC1 deletion drives RPE disorganization and loss of epithelial integrity

Histological evaluation of the ocular posterior segments of experimental animals 4 mo postinjection revealed regional RPE dysmorphia in AAV-Cre mice with enlarged and highly pigmented cells detached from the epithelial monolayer and surrounded by thick extracellular matrix deposition (Fig 6A). These regions of RPE degeneration were associated with marked photoreceptor thinning and loss of outer segments (OS) (Fig 6A). Ultrastructural analysis of the OS/RPE/choriocapillaris interface by TEM imaging revealed various stages of RPE phenotypic anomalies, characterized by severe loss of both apical microvilli and basal infoldings (Fig 6B and C); accumulation of large lipofuscin-filled granules (Fig 6C), melanosomes, and phagosomes with undigested OS (Fig 6D); thickening of the choriocapillaris endothelium and loss of fenestrations (Fig 6E); severe mitochondrial degeneration with swelling and loss of the cristae and incomplete outer-membrane closure characteristic of "mitochondria membrane ghosts" (Fig 6E).

## PGC-1 deletion promotes RPE EMT in vivo

Based on the morphological evidence of RPE dedifferentiation and our in vitro findings, we evaluated markers for oxidative phosphorylation, mesenchymal transition, and epithelial integrity

by immunofluorescence on cryosections. Evaluation of regions associated with overt retinal degeneration and RPE phenotypic changes showed decreased detection of the mitochondrial marker COX-IV after PGC-1 deletion, whereas the EMT markers—collagen-VI, vimentin, TWIST1, and ZEB1- were increased (Fig 7A). Interestingly, vimentin was also found bundled perinuclearly as in our in vitro model of PGC-1α silencing. Evaluation of global alteration in mitochondrial and EMT markers by staining and quantification of RPE/choroid flat mounts showed that whereas COX-IV was uniformly repressed in the RPE, EMT markers were only found increased in a subset of amorphic RPE cells (Fig S5). Investigation of the cytoskeletal and tight junction organization by phalloidin/DAPI staining on RPE/choroid flat mounts revealed in AAV-Cre mice a loss of hexagonal shape, increased multinucleation, and accumulation of large pigmented vacuoles (Fig 7B and C), a phenotype associated with age-related oxidative stress (Chen et al, 2016). Similar to our in vitro study, ZO-1 was found disorganized and absent in some regions (Fig 7B). Taken together, our results show that PGC-1α expression is required for the maintenance of RPE epithelial phenotype and function. The RPE dysfunction and mesenchymal transition caused by PGC-1α deletion is associated with disorganization of the outer retinal complex and severe retinal degeneration.

## Discussion

Our results show that PGC-1α is critical for the maintenance of RPE homeostatic processes, including glucose metabolism, autophagy,

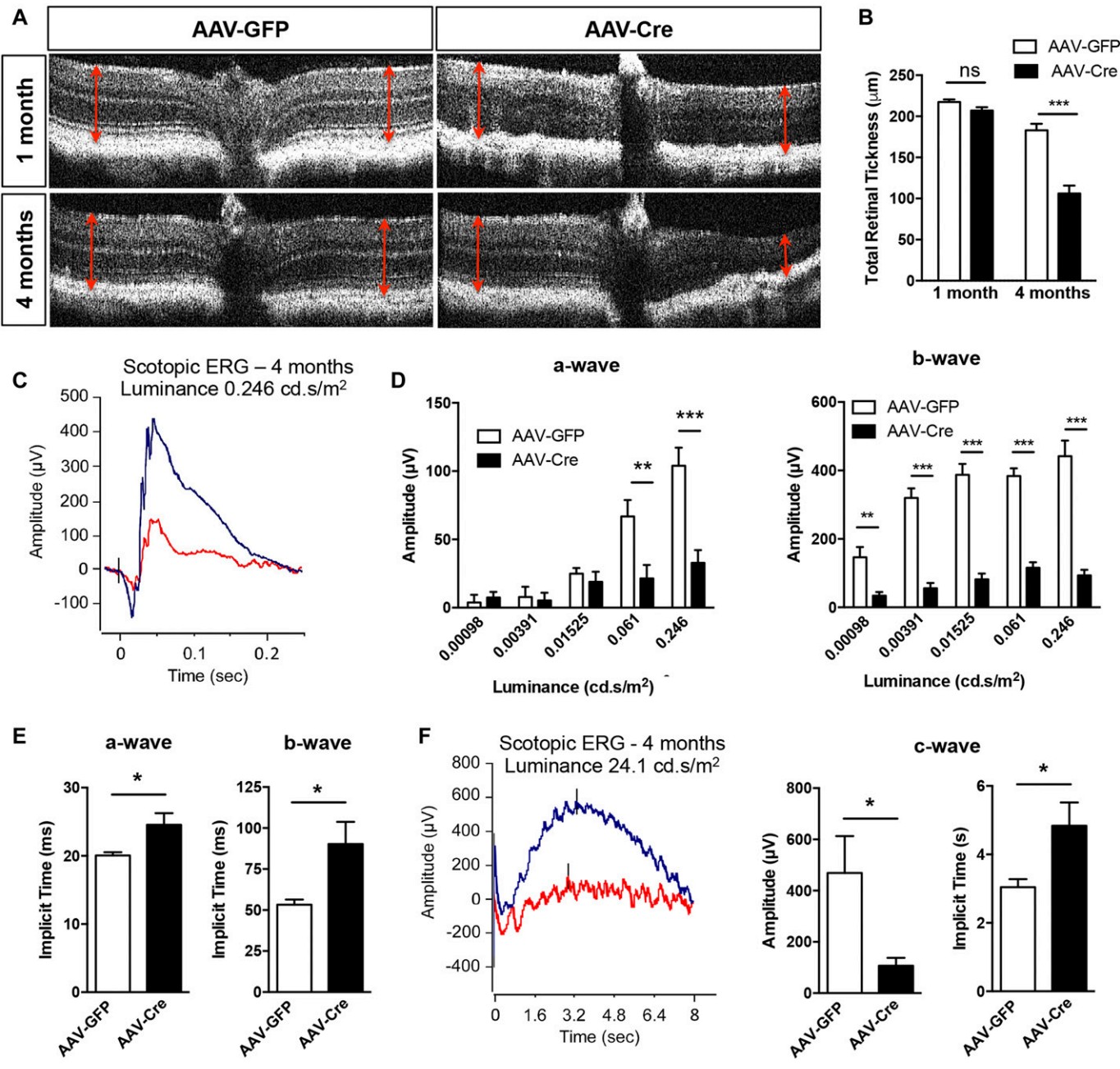

**Figure 5. Morphologic and functional evaluation of adult conditional PGC-1 knockout mice.**
**(A)** Representative OCT images from AAV-GFP and AAV-Cre–injected PGC-1A;B–floxed mice 1 and 4 mo postinduction. **(B)** Quantification of total retinal thickness from OCT imaging (n = 5–7). **(C)** Scotopic ERG recordings at a luminance of 0.246 (P) cd.s/m² from AAV-GFP (blue) and AAV-Cre (red) animals. **(D)** Quantification of *a*- and *b*-wave amplitudes for increasing stimulus intensity at 4 mo (n = 5). **(E)** Implicit times of *a*- and *b*-wave at light stimulus of 0.246 (P) cd.s/m² at 4 mo (n = 5). **(F)** RPE-generated *c*-wave amplitude and implicit time in AAV GFP group mice (blue) compared with AAV-GFP injected mice (red) measured with a light stimulus of 24.1 cd.s/m² at 4 mo (n = 6). Error bars are means ± SEM. Data were analyzed by unpaired *t* test. *P ≤ 0.05; **P ≤ 0.01; ***P ≤ 0.001 compared with their respective AAV-GFP controls.

and epithelial integrity. Our longitudinal analysis in vitro indicates that PGC-1α loss primarily alters RPE metabolic and autophagic functions followed by EMT induction. In vivo deletion of PGC-1α in adult RPE triggers loss of the epithelial phenotype and cellular disorganization associated with choroidal and neuronal degeneration. The phenotypic changes we observed in our mouse model of PGC-1α deletion in adult RPE are remarkably similar to that

described recently in aged NRF2/PGC-1α double knockout animals showing impaired autophagy, increased oxidative damage, and damaged mitochondria in RPE (Felszeghy et al, 2019). Interestingly, global NRF2/PGC-1α deficiency was also characterized by increased P62 and LC3B combined with accumulation of melanosomes, autolysosomes, and abnormal mitochondria (Felszeghy et al, 2019).

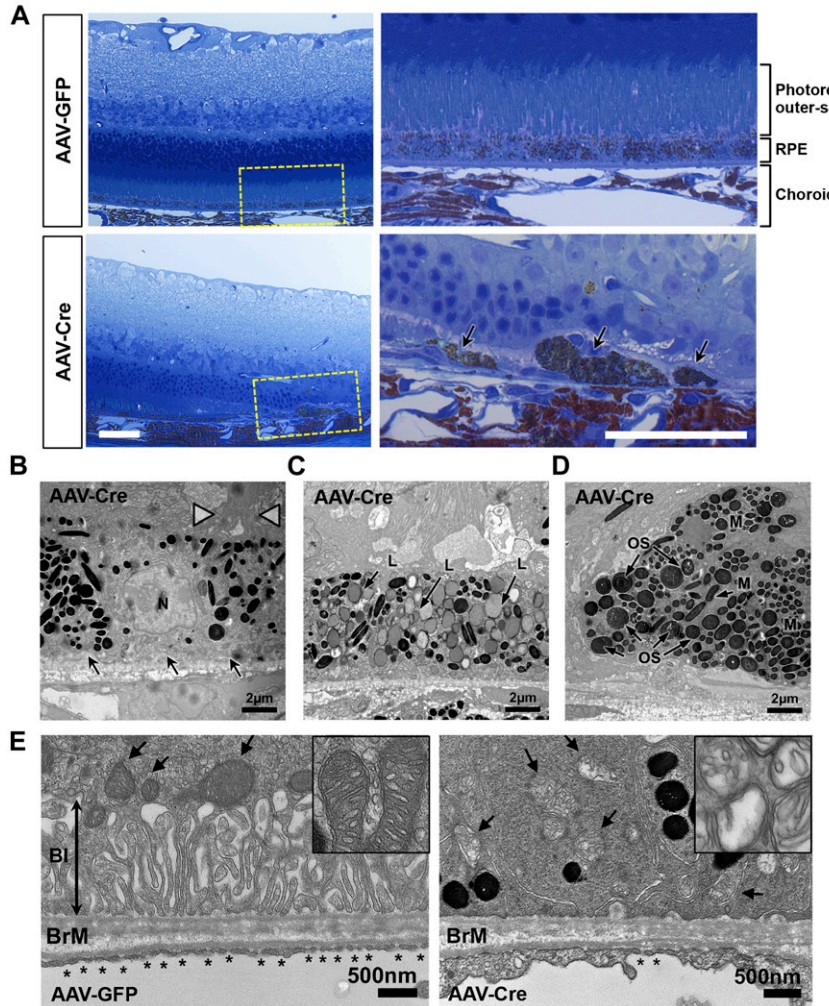

**Figure 6. PGC-1 deficiency triggers gross morphological changes in the outer retina and RPE.**
**(A)** Light micrographs from toluidine blue–stained semithin ocular sections from AAV-GFP and AAV-Cre–injected double floxed mice 4 mo postinjection showing severe disorganization of the RPE layer associated with prominent neuroretinal degeneration. Higher magnification of the RPE/choroid complex (corresponding to the dashed yellow boxes) shows degenerative lesion with swollen and sloughed RPE cells in the AAV-Cre–injected mice (arrows). Scale bars are 50 μm. **(B–D)** Transmission electron micrographs showing three representative examples of abnormal RPE in AAV-Cre–injected mice. Observed phenotypic changes vary from disorganized apical microvilli (arrowheads) with loss of basal infoldings (arrows) (B), massive accumulation of lipofuscin-containing granules (L) (C) to enlarged and dissociated cells filled with melanosomes (M) and undigested outer segments (OS) (D). Scale bars are 2 μm. **(E)** High magnification of the RPE basal side highlighting the prominent loss of basal infoldings (BI, double arrow), loss of fenestrations (*), thickening of the choriocapillaris endothelium, and severe mitochondria degeneration (arrows) marked by ruptured outer mitochondrial membranes and cristae remodeling (see insert). Scale bars are 500 nm.

In postmitotic cells, autophagy is the major mechanism for renewing damaged organelles and recycling nutrients. Mitochondrial health, antioxidant capacity, and autophagy are thereby tightly interconnected processes regulated by complex positive and negative feedback loops. Mitochondrial oxidative damage and/or decreased ATP levels can directly impair lysosomal activity (Demers-Lamarche et al, 2016), and defective lysosomal degradation can reciprocally cause the accumulation of defective mitochondria (Osellame et al, 2013). As PGC-1α transcriptionally induces numerous genes involved in all of these processes, teasing out the initial pathway(s) affected by PGC-1α silencing in RPE is difficult, but our longitudinal analysis offers interesting mechanistic cues. Our group has previously shown that PGC-1α is strongly induced during RPE functional maturation and that PGC-1α gain of function promotes (i) RPE metabolic function through induction of mitochondrial complex proteins and (ii) resistance to cytotoxic stress by inducing essential mitochondrial and cytoplasmic antioxidant enzymes (Iacovelli et al, 2016; Satish et al, 2018). Here, we show that, conversely, PGC-1α deficiency in RPE led to rapid loss of mitochondrial OXPHOS and reduced expression of catalase. These two early events may be key in promoting the increase in ROS

generation and autophagic dysfunction, which ultimately caused the loss of epithelial status in RPE. The rise in total ROS appeared to precede the detection of higher mitochondrial superoxide production which could be explained by an increase in mitochondrial calcium influx after mitochondrial stress (Ahmad et al, 2013) or reflect enhanced mitochondrial production of hydrogen peroxide and hydroxyl radical (Kirkland & Franklin, 2001).

PGC-1α is well known as a major transcriptional inducer of mitochondrial biogenesis. However, PGC-1α silencing in RPE did not alter the expression of the mitochondrial replication-related genes *TFAM* and *POLG* expression nor grossly reduce mitochondrial content. Even more unexpected was the detection of a robust increase in CS activity in shPGC-1α cells. Although CS activity is a well-accepted biomarker of mitochondrial mass and oxidative capacity, our findings suggest that CS activity may not be directly correlated with mitochondrial content in metabolically defective RPE cells. Indeed, mature and healthy RPE cells have been shown to preferentially use reductive carboxylation of α-ketoglutarate to produce citrate (Du et al, 2016) and induction of CS may, therefore, represent a compensatory mechanism for the cells to sustain citrate production (Mullen et al, 2014). Alternatively, increased CS

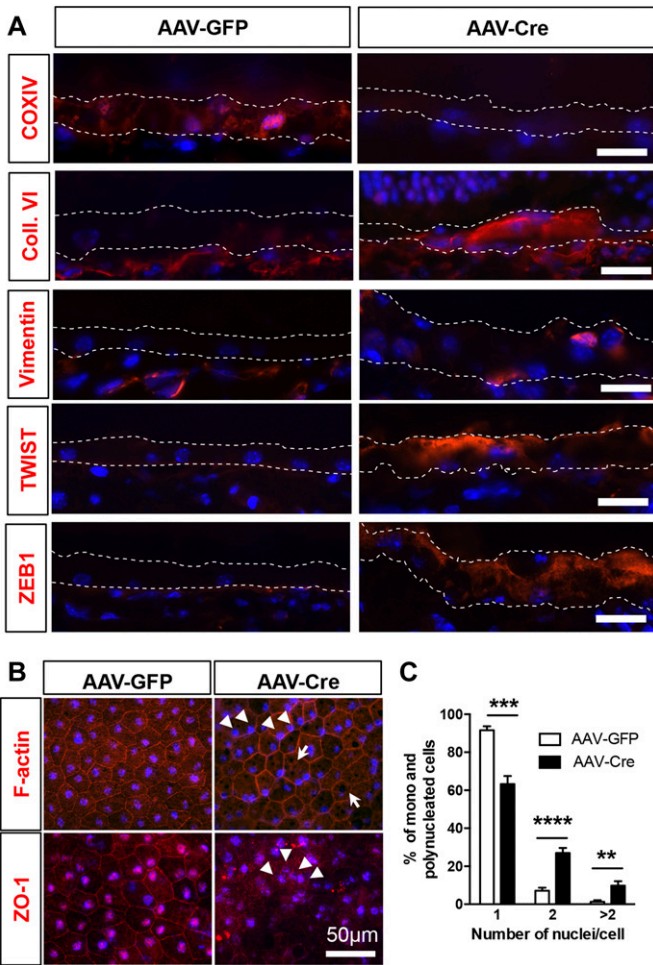

**Figure 7. RPE dedifferentiation and mesenchymal transition after PGC-1 deletion.**
**(A)** Immunodetection of mitochondrial and EMT markers on cryosections of eyes collected 4 mo postinjection with AAV-GFP or AAV-Cre, showing marked loss of the ECT component COXIV in the RPE layer (defined by the white dotted lines) of the AAV-Cre–injected animals, whereas the pro-EMT transcription factors TWIST1 and ZEB1 were strongly up-regulated. Mesenchymal phenotypic transition is further demonstrated by increased collagen VI deposition and vimentin perinuclear condensation. Scale bars are 20 $\mu m$. **(B)** RPE flat-mount preparation immunostained for F-actin or ZO-1 (red), demonstrating increased multinucleated cells (arrowheads) with large pigmented vacuoles (arrows) and loss of membranous ZO-1 distribution. Nuclei were costained with DAPI (Blue). Scale bar is 50 $\mu m$. **(C)** Percentage of mononucleated, binucleated, and polynucleated (>2 nuclei/cell) cells quantified on F-actin stained RPE flat mounts (n = 6). Error bars are means ± SEM. Data were analyzed by unpaired $t$ test. **$P \leq 0.01$; ***$P \leq 0.001$; ****$P \leq 0.0001$ compared with AAV-GFP group.

activity may be needed to support the biosynthetic requirements of PGC-1α–deficient cells by promoting citrate-derived fatty acid production and cell proliferation (MacPherson et al, 2017), which is consistent with the increased expression of PCNA at day 7.

Along with defective oxidative metabolism, the autophagic capacity of the RPE cells was severely altered after PGC-1α loss. Autophagic proteins are highly expressed in RPE cells and activation of the autophagic flux is essential for retinoid recycling and POS phagocytosis (Kim et al, 2013; Yao et al, 2014). As a direct target of the energy sensors and autophagy regulators AMPK and Sirt1,

PGC-1α is also able to directly promote autophagy by increasing the expression of the lysosomal transcription factor TFEB (Tsunemi et al, 2012). Similarly, we found numerous genes involved in autophagosome assembly to be induced after PGC-1α over-expression, whereas PGC-1α deficiency repressed their expression, thereby blunting the autophagic flux and promoting the accumulation of defective organelles. Importantly, our investigation of the faulty autophagic step in PGC-1α–deficient RPE cells point to the inability of AMPK to be activated upon starvation, suggesting the existence of a positive feedback mechanism between AMPK and PGC-1α regulating RPE metabolism and autophagic functions.

A key finding from our in vitro and in vivo models is that PGC-1α deficiency ultimately promoted the loss of epithelial characteristics in RPE and induced EMT conversion. Metabolic reprogramming is emerging as a critical driver for the mesenchymal engagement of tumor cells, but its involvement in EMT of RPE cells is unknown. RPE epithelial status is tightly controlled by homotypic cell–cell interactions and growth factor signaling. Thus, loss of cell contact and exposure to pro-EMT extracellular stimuli such as TGF-β or TNF-α have been proposed as the main molecular mechanisms for promoting EMT of RPE (Takahashi et al, 2010). Here, we observed that loss of PGC-1α function is sufficient to induce RPE mesenchymal conversion independently of any exogenous stimuli. Collectively, our data suggest mitochondrial and autophagic deficiency as the underlying mechanisms for EMT in RPE secondary to PGC-1α silencing. First, the early perinuclear condensation of the vimentin IFs that we detected is likely a direct consequence of increased ROS as spatial reorganization of the vimentin network can be rapidly induced by increased cytosolic ROS production via the activation of the GTPase-activating protein Cdc42GAP (Li et al, 2009). However, true mesenchymal engagement of the cells was only observed at a later stage with the induction of TWIST1 and ZEB1, suggesting that sustained mitochondrial/autophagic dysfunction is necessary for the up-regulation of EMT-transcriptional factors. Recent evidence indicates that glucose metabolites can exert a profound effect on gene expression by epigenetic modifications. In stem cells, aerobic glycolysis sustains pluripotency by promoting acetyl-CoA production from pyruvate and increasing histone acetylation (Moussaieff et al, 2015) whereas, in liver cells, selective histone acetylation from lipid-derived acetyl-CoA specifically up-regulates genes involved in lipid metabolism (McDonnell et al, 2016). This illustrates that acetyl-CoA–dependent gene regulation is contextually controlled as histone acetylation at selective loci depends on both the metabolic source and spatiotemporal availability of acetyl-CoA (Campbell & Wellen, 2018). In the context of PGC-1α deficiency, the metabolic conversion of RPE from OXPHOS to glycolysis along with increased CS activity may also augment the acetyl-CoA pool allowing for histone acetylation and the activation of EMT-associated genes. Consistent with this hypothesis, histone deacetylase inhibitors increased EMT markers, including ZEB1, TWIST1, and vimentin in prostate cancer cells (Kong et al, 2012). In addition to transcriptionally inducing EMT-promoting genes, PGC-1α loss may also augment EMT protein levels. Indeed, a direct consequence of chronic deficient autophagic flux in our experimental cells is the accumulation of the autophagy substrate p62 shown to interact with TWIST1 protein to promote its stabilization and tumor cell proliferation (Qiang et al, 2014). Together, these

findings indicate that PGC-1α–dependent regulation of glucose metabolism, metabolite shuttling, and autophagic flux in mature RPE cells is essential for the active repression of EMT and the maintenance of the cells' epithelial status under basal conditions.

RPE dedifferentiation and EMT is involved in the onset and progression of common blinding ocular diseases, including AMD and PVR. Similar to the phenotypes we observed after conditional PGC-1 deletion in adult RPE, RPE from dry AMD patients are characterized by degradation of mitochondria, accumulation of lysosomes and increased mesenchymal markers (Ferrington et al, 2016; Golestaneh et al, 2016, 2017; Ghosh et al, 2018). Although the direct involvement of PGC-1α in AMD pathogenesis remains to be determined, it is interesting to note that AMD-like RPE dysmorphia with reduced autophagic capacity and increased p62 was also reported in global heterozygous PGC-1α+/− mice (Zhang et al, 2018).

In many pathological conditions, visual impairment from retinal degeneration is a direct consequence of RPE metabolic dysfunction and dedifferentiation. This can be seen in our in vivo data showing that initial dysfunction of RPE after deletion of PGC-1s (as indicated by their impaired light-evoked response) leads to progressive damage to neighboring cells as shown by the markedly reduced photoreceptor and bipolar cell ERG responses, culminating in a dramatic thinning of the retinal layers. Abnormalities in the *c*-wave amplitude and implicit time indicate defects in ion conductance across the RPE apical and basal membranes (Wu et al, 2004) and thus, suggest a potential role for metabolic regulation of RPE transepithelial potentials, an interesting and under-explored area for future studies.

Considering EMT is a plastic and revertible process, there is therapeutic potential for manipulation. Thus, promoting the re-epithelialization of mesenchymal RPE in the early stage of the disease through metabolic rewiring is an attractive strategy as similar approaches are already being investigated for the treatment of aggressive tumors (Morandi et al, 2017). As a key regulator of the main biological processes supporting RPE oxidative metabolism and repressing mesenchymal conversion, PGC-1α is emerging as an interesting therapeutic target for ocular diseases associated with RPE dystrophy.

# Materials and Methods

### Cell culture and shRNA-mediated PGC-1α gene knockdown in RPE

The human RPE cell line, ARPE-19 (CRL-2302; ATCC), was expanded in DMEM/F12 medium with 2 mM L-glutamine supplemented with 10% penicillin/streptomycin (Lonza) and 10% FBS (Atlanta Biologicals) at 37°C, 10% $CO_2$. PGC-1α expression was silenced by lentivirus-mediated delivery of shRNAs selected from the Hannon-Ellege library (Silva et al, 2005) and cloned into a pGIPZ vector (Dharmacon). A pGIPZ nonsilencing shRNA (shControl) was used as negative control. To facilitate the delivery of shRNA into postmitotic cells, the pGIPZ-encoded shRNA was transfected into HEK293T cells along with a gag-pol-rev–containing packaging plasmid and an env-containing plasmid to pseudotype the viral capsids with the VSV-G (vesicular stomatitis virus glycoprotein) envelope protein

(Dharmacon) with TransIT 293 transfection reagent (Thermo Fisher Scientific). Stably transduced GFP+ cells were selected by FACS sorting. For all cell assays, cells were dissociated with trypsin EDTA (Lonza) and counted with a Beckman Coulter (Beckman Coulter Brea) before plating. For maturation studies, confluent shPGC1A and shControl ARPE-19 cells were differentiated in serum-free condition for up to 21 d. For AAS, confluent cells were washed three times in Hank's buffer (Gibco) and incubated for up to 24 h at 37°C in Earle's Balanced Salt Solution (EBSS; Sigma-Aldrich) (Martina et al, 2016). For autophagic flux analysis, the cells were pretreated for 4 h with 40 µM CQ (Sigma-Aldrich) before AAS with EBSS.

### Western blotting analysis

Protein lysates were prepared using 1× cell lysis buffer (Cell Signaling) supplemented with 1 mM PMSF (Sigma-Aldrich) and quantified using the bicinchoninic acid method (Thermo Fisher Scientific Pierce). Total cell lysate (30 to 80 µg) was loaded into 10% SDS polyacrylamide gel (Bio-Rad). Proteins were transferred to PVDF membranes (Millipore). The membranes were blocked in 5% nonfat milk in TBS-T or PBS plus 0.1% Tween (PBS-T) for 1 h at RT. Membranes were incubated with primary antibodies, including anti-PGC-1α (1:250, #ST1202; Millipore); anti-LC3 (1:1,000, #3868); anti-LC3B (1:1,000, #2775), anti-PCNA (1:1,000, #13110), anti-SQSTM1/p62 (1:1,000, #88588), anti-total (1:500, #2532) and phospho-AMPKα Thr172 (1:250, #2531), anti-phospho-S6 Ser235/236 (1:1,000, #2211), and anti-α-tubulin (1:1,000, #3873), all from Cell Signaling Technology; anti-apolipoprotein E (1:1,000, #AB947; Chemicon International); anti-GAPDH (1:1,000, #SC25778), anti-actin (1:1,000, #2Q1055), anti-Twist (1:250, #SC-81417), and anti-ZEB1 (1:250, #SC-515797), from Santa Cruz Biotechnology. Subsequently, the membranes were incubated with the appropriate secondary antibodies and developed by chemiluminescence with SuperSignal West Pico Chemiluminescent Substrate (Thermo Fisher Scientific Pierce) or fluorescence method LI-COR Odyssey (LI-COR). Equal loading and transfer were ascertained by reprobing the membranes for α--tubulin, GAPDH, and actin. Exposed films were scanned and the densitometry analyses were performed by Image J2 (Rueden et al, 2017) or using Image Studio 2.0 (LI-COR). In some assays, arbitrary units of fluorescence were converted to fold changes.

### Immunocytochemistry

ARPE-19 cells were seeded at a concentration of 50,000 cells/$cm^2$ on circle coverslips (12 mm of diameter). Once the cells reached 100% confluence, the complete medium was replaced by serum-free media for up to 21 d. Then, cells were washed with Hank's Balanced Salt Solution (Gibco), fixed with cold methanol (−20°C) for 10 min at RT and blocked with buffer solution (Hank's buffer, BSA 3% and 0.05% Tween) for 1 h. The cells were incubated with the appropriate primary antibodies: anti-ZO-1 (1:50, #61-7300; Invitrogen); anti-LAMP-1 (1:200, #9091; Cell Signaling Technology) and anti-Vimentin (1:200, #7783-500; Abcam), overnight at 4°C. The appropriate secondary antibodies were applied for 1 h at RT. Cell nuclei were stained with DAPI (Vector Laboratories INC). The slides were mounted with aqueous mounting medium (50% glycerol in PBS).

The images were acquired with an Axioscope microscope (Carl Zeiss Microscopy). For mitochondria staining, unfixed cells were exposed to 100 nM MitoTracker Orange CMTMRos (Life Technologies) in Hank's buffer for 30 min at 37°C, 10% $CO_2$. The ells were then washed, fixed with 4% paraformaldehyde in PBS, washed in Hank's buffer and permeabilized with 0.01% Triton X-100 (Sigma-Aldrich) for 5 min. Nuclei were stained using DAPI before mounting. Mitochondrial network images were acquired using an Axioscope microscope.

## Oxygen consumption rate and extracellular acidification rate measurement

Cells were plated at 50,000 cells/well in V7-PS microplates (Seahorse Bioscience) and matured for up to 21 d, For OCRs, the medium was replaced with the assay medium—minimal DMEM (Seahorse Bioscience) containing 2 mM glutamine (Lonza), 1 mM pyruvate (Gibco), and 25 mM glucose (Sigma-Aldrich), pH 7.4, and placed in a 37° $CO_2$-free incubator for 30 min. OCR was assessed using an XF-24 Extracellular Flux Analyzer (Seahorse Biosciences), at baseline and after the sequential addition of 2.5 $\mu$M oligomycin (Sigma-Aldrich) to inhibit complex V, 500 nM carbonyl cyanide-4-(trifluoromethoxy) phenylhydrazone (FCCP) (Seahorse Bioscience) to uncouple the proton gradient, 2 $\mu$M rotenone and 2 $\mu$M antimycin A (Sigma-Aldrich) to inhibit complex I and III, respectively (Iacovelli et al, 2016). All OCR ([% Base Line][pMoles/min]) values reported were corrected to the number of cells per well quantified by DAPI staining (converted to fold change) and expressed as percentage. For extracellular acidification rate (ECAR), the medium was replaced with the assay medium—minimal DMEM (Seahorse Bioscience) containing 1 mM glutamine (pH 7.4) and placed in a 37° C $CO_2$-free incubator for 1 h. ECAR was measured using an XF-24 Extracellular Flux Analyzer (Seahorse Biosciences) under basal conditions and after consecutive treatments with 10 mM glucose to induce glycolytic pathway, 2 $\mu$M oligomycin to promote glycolysis by blocking mitochondrial ATP production and 50 mM 2-deoxyglucose (2-DG) to block glycolysis. ECAR values (mpH/min) were normalized to the number of cells per well.

## ATP content and CSA

Total ATP levels were measured on cell lysates using the ATP bioluminescent assay CLS II kit (Roche) according to the manufacturer's instructions. Values for each sample were normalized to the protein concentration. CSA was quantified in fresh protein samples (60 $\mu$g/ml) using the MitoCheck Citrate Synthase Activity Assay Kit (Cayman Chemicals) according to the manufacturer's instructions.

## Quantification of mitochondrial superoxide generation and total reactive oxygen species

Cells (3 × 10$^5$) were harvested and stained with 5 $\mu$M MitoSOX (Molecular Probes) or 10 $\mu$M CM-$H_2$DCFDA (Life Technologies) at 37°C for 30 min to detect mitochondrial superoxide levels or total reactive oxygen species, respectively. The fluorescence intensity was analyzed using a fluorescence microplate reader (SynergyMx;

Biotek) at excitation and emission wavelengths of 530 and 590 nm and 485 and 528 nm, respectively. For each assay, the arbitrary units of fluorescence from 0.33 × 10$^5$ cells/well were averaged and converted to fold changes.

## Measurement of TER

Cells were seeded at 50,000 cells/well on laminin-coated 24-mm Transwell with 0.4-$\mu$m pore polyester membrane insert (Corning), left for 2 d to achieve confluency before maturation in serum-free medium for up to 21 d. TER was measured using an Epithelial Voltohmmeter (EVOM) with the STX100C electrode for 24-well format (World Precision Instruments). Resistance measurements in ohms ($\Omega$) were calculated by subtracting the resistance of the filter alone (background) from the values obtained with the filters with RPE cells and expressed in $\Omega/cm^2$ (Rosales et al, 2014).

## Adenoviral infection

ARPE-19 were infected with control adenovirus (Ad-GFP; Vector Biolabs) or with adenovirus containing the mouse PGC-1$\alpha$ sequence (Ad-mPGC-1$\alpha$, a gift of Dr. Arany, University of Pennsylvania, Philadelphia, PA) at a multiplicity of infection of 30 as previously described (Iacovelli et al, 2016). 24 h later, the cells were collected for RNA isolation. As PGC-1$\alpha$ functional domains are highly conserved across vertebrates (LeMoine et al, 2010), murine PGC-1$\alpha$ recapitulates the transcriptional functions of human PGC-1$\alpha$ (Diman et al, 2016).

## Animal studies and AAV injections

The animal work procedures complied with institutional animal care and authorizations in accordance with the local Committee for Ethics in Animal Research. Adult (8–12-wk old) C57Bl/6;PGC-1$\alpha^{(fl/fl)}$,PGC-1$\beta^{(fl/fl)}$ male and female mice (generously provided by Dr. Arany, University of Pennsylvania, USA) were anesthetized by IP injection of ketamine and xylazine. Mydriasis was induced by topical application of 1% tropicamide. Right eyes were injected subretinally with 1 $\mu$l (2 × 10$^9$ gc) of AAV2/8-CASI-GFP or AAV2/8-CASI-cre (as a mixture of 1.8 × 10$^9$ gc AAV2/8-cre + 2 × 10$^8$ gc AAV-GFP) using a 33G, point style three needle (Hamilton Company). 2 wk later, fundus imaging was performed to confirm transduction efficiency (GFP+) and lack of deleterious effects from the injection (subretinal hemorrhage, retinal detachment). Morphological and functional evaluations by fundus imaging, OCT, and ERG were performed at 1 and 4 mo postinjection. To control for potential Cre toxicity, 8-wk-old WT C57Bl/6 mice were subretinally injected with 2 × 10$^9$ AAV2/8-cre or AAV2/8-GFP and evaluated as the experimental cohort by OCT, fundus photography, and ERG.

## Fundus photography

Fundus images were obtained on live anesthetized animals using the Micron III retinal imaging system (Phoenix Research Laboratories). The images were acquired using the bright-field and autofluorescent mode (488 nm excitation, 500–700 nm emission).

## OCT and retinal thickness quantification

OCT images centered on the optic nerve head were acquired on live anesthetized animals using the Bioptigen OCT system (Bioptigen). For each animal, full retinal thicknesses were measured at precisely 405 $\mu$m from the center of optic nerve head and averaged from 8 clock positions (2–8, 12–6, 11–5, and 9–4), where 12 o'clock is superior and 3 o'clock is nasal.

## ERG

Full-field ERG recordings were measured in dark-adapted and anesthetized animals using the Diagnosys ColorDome Ganzfeld ERG system (Diagnosys LLC). Recordings of the $a$- and $b$-waves were performed at light intensities of 0.00098–0.246 (P) cd.s/m$^2$ and the $c$-wave response was measured as a flash intensity of 24.1 (P) cd.s/m$^2$. The analyses were performed using Espion V6 Diagnosys Software. For each recording, the $a$-wave amplitude and implicit time was measured from the baseline to the trough of the first negative wave; the $b$-wave amplitude was measured from the trough of the $a$-wave to the peak of the highest positive wave and the implicit time from the baseline up to the highest positive wave; the $c$-wave amplitude and implicit time was measured from the baseline to the maximum recorded peak.

## Histology and electron microscopy

At 4 mo postinjection, mice were deeply anesthetized by injection of ketamine (62.5 mg/kg) and xylazine (12.5 mg/kg). Live animals were perfused via the aorta with 10 ml of sodium cacodylate buffer (0.1 M, pH 7.4) followed by 10 ml of ½ Karnovsky's fixative in 0.1 M sodium cacodylate buffer (Electron Microscopy Sciences). Eyes were enucleated and the anterior segment removed. The eyecups were dehydrated and embedded in tEPON-812 epoxy resin. Semithin sections (1 $\mu$m) were stained with 1% toluidine blue in 1% sodium tetraborate aqueous solution for light microscopy. Ultrathin sections (80 nm) were cut from each sample block using a Leica EM UC7 ultramicrotome (Leica Microsystems) and stained with 2.5% aqueous gadolinium triacetate hydrate and Sato's lead citrate stains using a modified Hiraoka grid staining system (Seifert, 2017). Grids were imaged using an FEI Tecnai G2 Spirit transmission electron microscope (FEI Company) at 80 kV interfaced with an AMT XR41 digital CCD camera (Advanced Microscopy Techniques) for digital TIFF file image acquisition. TEM imaging of retina samples were assessed and digital images captured at 2k×2k pixel, 16-bit resolution.

## Immunohistochemistry

Eyes were enucleated and fixed in 4% paraformaldehyde at 4°C overnight. RPE/choroid flat mounts were prepared by removal of the anterior segment and the retina. For cryosections, eyecups were prepared by removing the anterior segment and incubated in 30% sucrose overnight before embedding in Tissue-Tek OCT (Sakura Finetek) and cryosectioning (10 $\mu$m). RPE/choroid preparations and cryosections were blocked in 10% BSA, Triton 0.5% in PBS at RT for at least 1 h. Antimouse IgG fragment (0.12 mg/ml, #115-007-003; Jackson ImmunoResearch Laboratories) was added to the blocking buffer when using a mouse-raised primary antibody. The samples were then incubated overnight at 4°C with the following primary antibodies in blocking buffer: rabbit anti-Cre (1:100, #NB100-56133SS; Novus Biologicals), anti-GFP (1:200, #A-11122; Thermo Fisher Scientific), anti-ZO-1 (1:50, #61-7300) and Alexa 594–conjugated Phalloidin (1:100, #A-12381) both from Thermo/Invitrogen, rabbit anti-COX IV (1:200, #ab16056) and anti-Collagen VI (1:200, #ab6588) from Abcam; rabbit anti-Vimentin (1:200, #5741, Cell Signalling), and mouse anti-TWIST (1:250, #SC-81417) and ZEB-1 (1:250, #SC-515797) from Santa Cruz Biotechnology. After washing with PBS, the samples were incubated with a DyLight 549 goat anti-rabbit antibody (1:500; Vector Laboratories INC) or Alexa 594 anti-mouse antibody (1:500, #11062; Invitrogen) 1 h at RT. Cell nuclei were stained with DAPI (Vector). The slides were mounted with aqueous mounting medium (50% glycerol in PBS). The images were acquired with an Axioscope microscope (Carl Zeiss Microscopy). For quantification of mean fluorescence intensity, four to six randomly selected fields with flat RPE layout were imaged for each sample using identical exposure time. From each image, the median pixel intensity in the red channel was quantified using Adobe Photoshop CS6 software, and the results were averaged per sample.

## Real-time quantitative PCR analysis

Total mRNA from ARPE-19, RPE/choroid, and retina was isolated using RNA-Bee solution (IsoText Diagnostic Inc.). RNA was reverse-transcribed using the iScript kit (Bio-Rad). qPCR reactions were performed using the SYBR Green Master mix and the ABI Prism 9700 Sequence Detection System (Applied Biosystems) according to the manufacturer's instructions, using specific primers (Table 1). Ct values were normalized to the housekeeping genes, *PPIA* and *HRPT1*, and relative changes in gene expression were calculated using the ΔΔCt method.

## Statistical analysis

Appropriate statistical tests were applied using GraphPad Prism 5.0 (GraphPad). In brief, $t$ tests were used for comparisons between two groups. One-way ANOVA with Tukey post hoc test was used for comparison of three or more groups with one independent variable. Values were expressed as mean ± SEM. Significance was considered when $P \leq 0.05$ and is indicated in the text as follows: *$P$ < 0.05, **$P$ < 0.01, ***$P$ < 0.001, and ****$P$ < 0.0001.

# Supplementary Information

# Acknowledgements

This work was supported by the Alcon Research Institute, Young Investigator Research Grant (M Saint-Geniez); the Research to Prevent Blindness Dolly Green Special Scholar Award (M Saint-Geniez); the Grimshaw-Gudewicz Charitable Foundation (M Saint-Geniez); the Iraty Award (M Saint-Geniez); CNPq, National Council of Scientific and Technological Development—Brazil (PDE 210474/2014-9) (MAB Rosales); and the NEI Core Grant P30EYE003790.

**Table 1. Primer sequences for qPCR.**

| Gene symbol | Gene name | Forward sequence (5′–3′) | Reverse sequence (5′–3′) |
|---|---|---|---|
| Human primers | | | |
| ATG4D | Autophagy related 4D cysteine peptidase | CTCAACCCCGTGTATGTGC | TACAGTGAGTGTCGCGGTTT |
| ATG9B | Autophagy related 9B | GCTACTGGGACATCCAGGTG | AAGAGGCGGGACTGCAC |
| CAT | Catalase | ACTTTGAGGTCACACATGACATT | CTGAACCCGATTCTCCAGCA |
| FIS1 | Mitochondrial fission 1 protein | TGACATCCGTAAAGGCATCG | CTTCTCGTATTCCTTGAGCCG |
| GPX | Glutathione peroxidase 1 | CCAGTCGGTGTATGCCTTCTC | GAGGGACGCCACATTCTCG |
| HMOX1 | Heme oxygenase 1 | GCCAGCAACAAAGTGCAAG | GAGTGTAAGGACCCATCGGA |
| HPRT1 | Hypoxanthine phosphoribosyltransferase 1 | CCTGGCGTCGTGATTAGTGAT | AGACGTTCAGTCCTGTCCATAA |
| LAMP1 | Lysosomal-associated membrane protein 1 | ACGTTACAGCGTCCAGCTCAT | TCTTTGGAGCTCGCATTGG |
| MAP1LC3B | Microtubule associated protein 1 light chain 3 beta | GAGAAGACCTTCAAGCAGCG | TATCACCGGGATTTTGGTTG |
| MCOLN1 | Mucolipin 1 | TTGCTCTCTGCCAGCGGTACTA | GCAGTCAGTAACCACCATCGGA |
| MFN2 | Mitofusin 2 | ATGTGGCCCAACTCTAAGTG | CACAAACACATCAGCATCCAG |
| PPARGC1A | Peroxisome proliferator-activated receptor gamma, coactivator 1 alpha | AGCCTCTTTGCCCAGATCTT | CTGATTGGTCACTGCACCAC |
| PPARGC1B | Peroxisome proliferator-activated receptor gamma, coactivator 1 beta | CCACATCCTACCCAACATCAAG | CACAAGGCCGTTGACTTTTAGA |
| POLG | DNA polymerase gamma, catalytic subunit | GAAGGACATTCGTGAGAACTTCC | GTGGGGACACCTCTCCAAG |
| PPIA | Peptidylprolyl isomerase A | CAGACAAGGTCCCAAAGACAG | TTGCCATCCAACCACTCAGTC |
| PRC | PPARG related coactivator 1 (PPRC1) | GTGGTTGGGGAAGTCGAAG | TGACAAAGCCAGAATCACCC |
| SOD1 | Superoxide dismutase 1, soluble | AGGGCATCATCAATTTCGAGC | GCCCACCGTGTTTTCTGGA |
| SOD2 | Superoxide dismutase 2, mitochondrial | CAGACCTGCCTTACGACTATGG | CGTTCAGGTTGTTCACGTAGG |
| SIRT1 | Sirtuin 1 | TCAGTGTCATGGTTCCTTTGC | AATCTGCTCCTTTGCCACTCT |
| SIRT3 | Sirtuin 3 | TGGAAAGCCTAGTGGAGCTTCTGGG | TGGGGGCAGCCATCATCCTATTTGT |
| TFAM | Transcription factor a, mitochondrial | CCATATTTAAAGCTCAGAACCCAG | CTCCGCCCTATAAGCATCTTG |
| TP53 | Tumor protein p53 | TCAACAAGATGTTTTGCCAACTG | ATGTGCTGTGACTGCTTGTAGATG |
| TXN2 | Thioredoxin 2 | TGATGACCACACAGACCTCG | ATCCTTGATGCCCACAAACT |
| TWIST1 | Twist family BHLH transcription factor 1 | CGGAGAAGCTGAGCAAGATT | TGGAGGACCTGGTAGAGGAA |
| VIM | Vimentin | GAGAACTTTGCCGTTGAAGC | TCCAGCAGCTTCCTGTAGGT-3 |
| WIPI1 | WD repeat domain, phosphoinositide interacting 1 | CCTCCTGGATATTCCTGCAA | GCACAATCTCCCCTGAAGTC |
| ZEB1 | Zinc finger E-box binding homeobox 1 | AGTGATCCAGCCAAATGGAA | TTTTTGGGCGGTGTAGAATC |
| ZEB2 | Zinc finger E-box binding homeobox 2 | AACAAGCCAATCCCAGGAG | GTTGGCAATACCGTCATCCT |
| Mouse primers | | | |
| Hprt1 | Hypoxanthine phosphoribosyltransferase 1 | TCAGTCAACGGGGGACATAAA | GGGGCTGTACTGCTTAACCAG |
| Ppargc1a | Peroxisome proliferator-activated receptor gamma, coactivator 1 alpha | AGCCGTGACCACTGACAACGAG | GTCGCATGGTTCTGAGTGCTAAG |
| Ppargc1b | Peroxisome proliferator-activated receptor gamma, coactivator 1 beta | CCCAGCGTCTGACGTGGACGAGC | CCTTCAGAGCGTCAGAGCTTGCTG |
| Ppia | Peptidylprolyl isomerase A | GAGCTGTTTGCAGACAAAGTTC | CCCTGGCACATGAATCCTGG |

## Author Contributions

MAB Rosales: conceptualization, data curation, formal analysis, investigation, methodology, and writing—original draft, review, and editing.
DY Shu: data curation, formal analysis, investigation, methodology, and writing—original draft, writing, review, and editing.
J Iacovelli: formal analysis, investigation, and writing—review and editing.

M Saint-Geniez: conceptualization, data curation, formal analysis, supervision, funding acquisition, investigation, methodology, project administration, and writing—original draft, review, and editing.

## Conflict of Interest Statement

The authors declare that they have no conflict of interest.

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
