## [Reviewer comments · Life Science Alliance]

Life Science Alliance

Loss of PGC-1 α in RPE induces mesenchymal transition and promotes retinal degeneration

Mariana Aparecida Rosales, Daisy Shu, Jared Iacovelli, and Magali Saint-Geniez

DOI: <https://doi.org/10.26508/lsa.201800212>

Corresponding author(s): Magali Saint-Geniez, Schepens Eye Research Institute of Massachusetts Eye and Ear

Review Timeline:

Submission Date:	2018-10-10
Editorial Decision:	2018-11-21
Revision Received:	2019-04-08
Editorial Decision:	2019-04-30
Revision Received:	2019-05-07
Accepted:	2019-05-09

Scientific Editor: Andrea Leibfried

Transaction Report:

November 21, 2018

Re: Life Science Alliance manuscript #LSA-2018-00212-T

Dr. Magali Saint-Geniez
Schepens Eye Research Institute
Department of Ophthalmology, Harvard Medical School
20 Staniford St
Boston, MASSACHUSETTS 02114

Dear Dr. Saint-Geniez,

Thank you for submitting your manuscript entitled "Loss of PGC-1 α in RPE induces mesenchymal transition and promotes retinal degeneration" to Life Science Alliance. The manuscript was assessed by expert reviewers, whose comments are appended to this letter.

As you will see, the reviewers appreciate your data and they provide constructive input on how to further strengthen your manuscript. The individual points raised seem all straightforward to address with a minor revision, and we would thus like to invite you to provide such a revised manuscript. Please note that the requested further reaching insight for PGC1a binding to promoters (ChIP-seq analysis; reviewer #1) is not needed for acceptance here. Please let us know in case you would like to discuss individual revision points further. Many of them seem to be addressable by changes to the text/discussion and by re-analysis of data already at hand, and we are happy to provide further guidance for this once you have had the time to consider the individual concerns.

Thank you for this interesting contribution to Life Science Alliance. We are looking forward to receiving your revised manuscript.

Sincerely,

- A letter addressing the reviewers' comments point by point.
- An editable version of the final text (.DOC or .DOCX) is needed for copyediting (no PDFs).
- High-resolution figure, supplementary figure and video files uploaded as individual files: See our detailed guidelines for preparing your production-ready images, <http://life-science-alliance.org/authorguide>
- Summary blurb (enter in submission system): A short text summarizing in a single sentence the study (max. 200 characters including spaces). This text is used in conjunction with the titles of papers, hence should be informative and complementary to the title and running title. It should describe the context and significance of the findings for a general readership; it should be written in the present tense and refer to the work in the third person. Author names should not be mentioned.

B. MANUSCRIPT ORGANIZATION AND FORMATTING:

Full guidelines are available on our Instructions for Authors page, <http://life-science-alliance.org/authorguide>

Reviewer #1 (Comments to the Authors (Required)):

The authors show that markedly reduced PGC1a expression in RPE cells causes mitochondrial dysfunction and defective autophagy that is associated with loss of epithelial phenotype in vitro. They follow up these findings in vivo using AAV delivered Cre to delete PGC1a and PGC1b in the

RPE of mice that results in disorganization of the RPE and loss of apparently functional mitochondria (based on EM). Similar to results obtained in vitro, this was associated with loss of epithelial phenotype and in vivo resulted in retinal degeneration.

1. The authors show decreased expression of autophagy genes upon PGC1a knockdown and increased expression with PGC1a over-expression. They also show defective LC3 processing in PGC1a knockdown cells. However to confirm that toggling PGC1a is actually inhibiting autophagy as opposed to increasing the rate of autophagic flux, the authors need to repeat experiments shown in figure 2E and 2F in the presence or absence of bafilomycin A1 or chloroquine to determine whether LC3B-II now accumulates or not.
2. Work in figure 3 overstates its conclusions. While knocking down PGC1a appears to cause mitochondrial dysfunction and defective autophagy, and also loss of epithelial phenotype, the authors cannot say that the loss of epithelial phenotype is due to mitochondrial dysfunction and/or defective autophagy. This is merely correlative.
3. It is not at all clear from figure 7A, how many cells are present and this figure needs to be analyzed quantitatively. Nor is it clear why the format of analyzing expression in figure 7B is not used for the other markers also (COXIV, Twist, Vimentin etc).

This is an interesting and provocative set of preliminary observations but currently much of what is presented is descriptive. Identifying a novel role for PGC1a in autophagy and/or EMT would be particularly significant. However, at the moment these findings are not sufficiently fleshed out and the connection between defective mitochondria/autophagy and EMT is not at all clear. The author needs to determine whether key autophagy genes shown to be down-regulated in the RPE cells are direct PGC1a targets - perform some ChIP to assess whether PGC1a binds the promoters of these genes. They also need to explain why mitochondria are dysfunctional when PGC1a is down-regulated - due to defective mitophagy or is it due to reduced biogenesis? How does the mitochondrial defect relate to the impact on EMT with loss of epithelial phenotype? Overall more work is needed to "drill down" on each of these findings to make the work sufficiently compelling for publication.

Reviewer #2 (Comments to the Authors (Required)):

Mariana Aparecida Brunini Rosales et al show that PGC-1 α depletion leads to impaired autophagy and epithelial-mesenchymal transition in RPE cells resembling age-related macular degeneration (AMD). This is excellent manuscript. I liked very much to read it. Hypothesis is very relevant. Study is very well planned. Data is strong and convincing. Manuscript is very well written. Discoveries may really open new ways to find better treatments to prevent or treat AMD. I have only minor comments to improve manuscript.

1. In Fig. 6A-D indicate organelle or tissue changes by arrow.
2. Autophagosomes and autolysosomes were not calculated to confirm autophagy flux by TEM, but this can be replaced by citing and discussing the recent publication: Loss of NRF-2 and PGC-1 α genes leads to retinal pigment epithelium damage resembling dry age-related macular degeneration. Felszeghy S, Viiri J, Paterno JJ, Hyttinen JMT, Koskela A, Chen M, Leinonen H, Tanila H, Kivinen N, Koistinen A, Toropainen E, Amadio M, Smedowski A, Reinisalo M, Winiarczyk M,

Mackiewicz J, Mutikainen M, Ruotsalainen AK, Kettunen M, Jokivarsi K, Sinha D, Kinnunen K, Petrovski G, Blasiak J, Bjørkøy G, Koskelainen A, Skottman H, Urtti A, Salminen A, Kannan R, Ferrington DA, Xu H, Levonen AL, Tavi P, Kauppinen A, Kaarniranta K. *Redox Biol.* 2019;20:1-12. doi: 10.1016/j.redox.2018.09.011

Reviewer #3 (Comments to the Authors (Required)):

The authors of the manuscript titled 'Loss of PGC-1 α in RPE induces mesenchymal transition and promotes retinal degeneration' have undertaken a thoroughly planned, detailed investigation on the consequences of PGC-1 α functional loss in retinal pigment cells. In 2016 they had published an article describing the role of PGC1a with respect to influencing oxidative metabolism and antioxidant capacity in ARPE19 cells. In the current study the group analysed loss of PGC-1 α in RPE and genetically modified mice (C57Bl/6;PGC-1 α (fl/fl),PGC-1 β (fl/fl) conditionally knocking both forms out in RPE via CRE activity.

In the current manuscript, the authors convincingly show mitochondrial dysfunction and oxidative damage after repressing PGC-1 α expression. Knock-down/silencing in RPE cells resulted in epithelial dedifferentiation and mesenchymal transition. Moving from isolated RPE cells to adult mice, they can show, that conditional knockout of PGC-1 coactivators resulted in a rapid RPE dysfunction and transdifferentiation in vivo associated with severe photoreceptor degeneration.

The study builds on thoroughly generated, reproduced data. Data are presented in a comprehensive and clear way. References are provided in a complete and meaningful way. Workplan and the resulting data are conceivable and convincing. Each figure is designed such, that it complies with the experimental plan of the study. The results are well discussed in the context of the current level of understanding.

Taken together, this is a convincing manuscript that merits publication.

However, I would encourage the authors to consider the following suggestions concerning further improvement of the manuscript:

As human RPE cells as well mice are used, the coverage between mouse sequence and human sequence of PGC1a (line 495) in the overexpression experiment would be of interest.

EMT can be associated with reduced proliferation (i.e. some forms of cancer). I wonder whether cell numbers change after PGC1a silencing, I would have expected no differences in this case, because the energy metabolism switch and autophagy already require great effort for RPE cells. This is addressed in fig 3: cell numbers increased. However, this is no further investigated or discussed.

Figure 1: PGC-1 α repression causes RPE mitochondrial dysfunction and oxidative stress
Fig 1E) CSA activity increases at day 14, then decreases at 21 days for shPGC1A. Please discuss!
Fig 1H) Total ROS activity appears at a maximum at day 7 as comparing to all other timepoints. What's the reason for that? (a point for discussion)
Fig 1I) The authors point to downregulated genes, how about upregulation of HMOX1 and TXN2 genes at day 7 and day 14?

Figure 2; Line 142: PGC-1 α silencing impairs RPE autophagic flux

Figure 2B) It appears more suitable to adjust the order of the relative expression of genes within the graphs to the text as;

LAMP1, MAP1LC3B, WIPI, ATG4D, ATG9B

Figure 2E) Why are there two different western blotting bands for AAS and untreated conditions?

To show two different repeats?

Figure 2H) The label for " Untreated versus AAS" similar to graphs Figure 2 E and Fig 2F is missing (Headline).

Figure 3 E, F, G, H) -> For western blotting experiments GAPDH bands are not equal (lower in the treated panel). Did the authors check for equal protein concentration?

Figure 5 D and F) "UV" should be changed to "V" (ERG graph)

Figure 7 A) Mitochondrial/autophagic dysfunction and oxidative damage is shown in vitro. The authors show markers for oxidative phosphorylation and mesenchymal transition in vivo. How about any data to show autophagic dysfunction based on in vivo data?

For the discussion part:

- Discussion on the age dependency of degenerating mouse retina is given for the AAV-cre mice. Discussion on the age dependency of results of RPE silenced for PGC-1 α , however, is missing.

Based on the in vitro data using 7, 14 and 21 days, this appears attractive.

- Functional data (ERG) could be discussed deeper in the discussion part? How do functional and histological data relate?

- With respect to Iba-1 as a marker revealing an immune response, is there a pattern?

- Does photoreceptor morphology change upon PGC-1 α silencing?

- PGC-1 α has a role in serving as a switch between mitochondrial biogenesis and oxidative damage by controlling the mitochondrial levels of ROS (Kaarniranta et al., 2018). What do the authors conclude with respect of a potential switch mechanism and effect on the mesenchymal transition?

- Line 278 The authors state: "Our longitudinal analysis in vitro indicates that PGC-1 α loss alters primarily 279 RPE metabolic and autophagic functions followed by EMT induction."

How do these patterns as a consequence of loss of PGC-1 α in retinal pigment epithelial (RPE) cells trigger mitochondrial/autophagic dysfunction and oxidative damage mechanistically?

- Line 285 Mitochondrial oxidative damage and/or decreased ATP levels can directly impair lysosomal activity (Demers-Lamarche et al., 2016) and defective lysosomal degradation can reciprocally cause the accumulation of defective mitochondria (Osellame et al., 2013)

Did the authors check ATP level of the cells after silencing PGC-1 α ?

- Line 295 The rise in total ROS appeared to precede the detection of higher mitochondrial superoxide production which could be explained by an increase in mitochondrial calcium influx following mitochondrial stress (Ahmad et al., 2013) or reflect enhanced mitochondrial production of hydrogen peroxide and hydroxyl radical, both detectable with the probe used.

Did the authors check for mitochondrial calcium influx and/or hydrogen peroxide and hydroxyl radical or is it an assumption?

- Line 301 PGC-1 α is well known as a major transcriptional inducer of mitochondrial biogenesis. However, PGC-1 α silencing in RPE did not alter the expression of the mitochondrial replication-related genes TFAM and POLG expression or grossly reduced mitochondrial content.

If mitochondrial protein composition is not severely affected, how can loss of mitochondrial dysfunction be mechanistically explained?

- Line 324 ...autophagic flux and promoting the accumulation of defective organelles. Importantly our investigation of the faulty autophagic step in PGC-1 α -deficient RPE cells point to the inability of AMPK to be activated upon starvation suggesting the existence of a positive feedback mechanism

between AMPK and PGC-1 α regulating RPE metabolism and autophagic functions.
Is there any reference supporting this conclusion?

Reviewer #1:

“The authors show that markedly reduced PGC1a expression in RPE cells causes mitochondrial dysfunction and defective autophagy that is associated with loss of epithelial phenotype in vitro. They follow up these findings in vivo using AAV delivered Cre to delete PGC1a and PGC1b in the RPE of mice that results in disorganization of the RPE and loss of apparently functional mitochondria (based on EM). Similar to results obtained in vitro, this was associated with loss of epithelial phenotype and in vivo resulted in retinal degeneration.”

We thank the reviewer for his/her thorough reading of our manuscript and detailed comments on how it could be improved.

Concern 1: “The authors show decreased expression of autophagy genes upon PGC1a knockdown and increased expression with PGC1a over-expression. They also show defective LC3 processing in PGC1a knockdown cells. However to confirm that toggling PGC1a is actually inhibiting autophagy as opposed to increasing the rate of autophagic flux, the authors need to repeat experiments shown in figure 2E and 2F in the presence or absence of bafilomycin A1 or chloroquine to determine whether LC3B-II now accumulates or not. “

Answer: We agree with the reviewer’s comment. As suggested we have repeated the starvation experiments in presence of the lysosomal inhibitor chloroquine and monitored the autophagic flux by quantification of LC3-II levels. As expected, we found that LC3-II levels increased in shCtrl cells following both serum starvation and chloroquine treatments, indicating enhancement of the autophagic flux (Mizushima et al., 2010). In contrast, LC3-II levels were unchanged in PGC-1 α silenced cells following both serum starvation and/or chloroquine treatment (New Figure S2B). Thus, our results suggest that silencing PGC-1 α impairs RPE’s autophagic flux and further highlight the importance of PGC-1 α in promoting autophagy.

Reference:

Mizushima, N., T. Yoshimori, and B. Levine. 2010. Methods in Mammalian Autophagy Research. *Cell*.

Concern 2: “Work in figure 3 overstates its conclusions. While knocking down PGC1a appears to cause mitochondrial dysfunction and defective autophagy, and also loss of epithelial

phenotype, the authors cannot say that the loss of epithelial phenotype is due to mitochondrial dysfunction and/or defective autophagy. This is merely correlative.”

Answer: We thank the reviewer for his/her input. We agree that our data does not provide definitive evidence that the Oxphos and autophagic dysfunctions following PGC-1 α loss are directly causative of the RPE dedifferentiation and EMT observed, as these biological processes are tightly linked and interdependent. Teasing out the hierarchic organization between metabolic, autophagic and phenotypic pathways is thereby highly challenging. Further investigation of a direct contribution of mitochondrial dysfunction and/or glycolytic switch to EMT of RPE cells (independently of PGC-1 α) will be particularly valuable. However our central premise was to define the role of PGC-1 α on RPE metabolic and functional maturation. Our conclusions are further supported by our longitudinal analysis indicative of a rapid metabolic dysfunction (by day 7) preceding detectable induction of the mesenchymal transcription factors Zeb1, Twist1 and repression of p53 at day 21. As suggested we have adjusted the conclusions drawn from Figure 3 as “Taken together, these data indicate that PGC-1 α is required to maintain RPE epithelial phenotype and that sustained PGC-1 α loss triggers EMT in RPE.” (Lines 216-217 page 10).

Concern 3: “It is not at all clear from figure 7A, how many cells are present and this figure needs to be analyzed quantitatively. Nor is it clear why the format of analyzing expression in figure 7B is not used for the other markers also (COXIV, Twist, Vimentin etc).”

Answer: Sections shown in Figure 7 are representative of selected regions associated with overt retinal degeneration and RPE phenotypic changes. In these locations we observed increased EMT-markers in highly amorphic RPE. As described in Figure 6, RPE degeneration caused by PGC-1s deletion is regional (Fig 6A) and the phenotypic changes ranged from minor to severe (Fig 6B-E). In order to provide additional information on the global alteration in mitochondrial and EMT-marker expression we stained and quantified these proteins on RPE flat-mounted preparations as recommended. The new figure S5 shows that while the mitochondrial protein CoxIV was uniformly repressed in RPE, EMT markers were only found increased in a regional subset of amorphic RPE cells.

Concern 4: “The author needs to determine whether key autophagy genes shown to be down-regulated in the RPE cells are direct PGC1 α targets - perform some ChIP to assess whether PGC1 α binds the promoters of these genes.”

Answer: As a transcriptional co-factor, PGC-1 α does not directly bind to DNA sequences but forms a complex with many transcription factors (TFs) in order to co-activate a large and intricate transcriptional network. Genome-wide analysis of the binding sites for PGC-1 α -TFs complexes has identified numerous transcription factor targets including CEBPB, ERR α , GABP (Charos et al., 2012; Chang et al., 2018) known to transcriptionally promote autophagy- and lysosome-related genes (Guo et al., 2013; Kim et al., 2018; Zhu et al., 2014). Our findings, based on gain and loss experiments and showing that PGC-1 α transcriptionally regulates multiple autophagy-associated genes in RPE cells, are consistent with previous works in other cells and tissues (Vainshtein et al., 2015; Tsunemi et al., 2012).

References:

- Chang, J.S., S. Ghosh, S. Newman, and J.M. Salbaum. 2018. A map of the PGC-1 α - and NT- PGC-1 α -regulated transcriptional network in brown adipose tissue. *Sci. Rep.* 8:277. doi:10.1038/nprot.2008.211.
- Charos, A.E., B.D. Reed, D. Raha, A.M. Szekely, S.M. Weissman, and M. Snyder. 2012. A highly integrated and complex PARGC1A transcription factor binding network in HepG2 cells. *Genome Res.* 22:1668–1679. doi:10.1101/gr.127761.111.
- Guo, L., J.-X. Huang, Y. Liu, X. Li, S.-R. Zhou, S.-W. Qian, Y. Liu, H. Zhu, H.-Y. Huang, Y.-J. Dang, and Q.-Q. Tang. 2013. Transactivation of Atg4b by C/EBP β Promotes Autophagy To Facilitate Adipogenesis. *Mol Cell Biol.* 33:3180–3190. doi:10.1172/JCI42601.
- Kim, S.Y., C.-S. Yang, H.-M. Lee, J.K. Kim, Y.-S. Kim, Y.-R. Kim, J.-S. Kim, T.S. Kim, J.-M. Yuk, C.R. Dufour, S.-H. Lee, J.-M. Kim, H.-S. Choi, V. Giguère, and E.-K. Jo. 2018. ESRRA (estrogen-related receptor. 1–18. doi:10.1080/15548627.2017.1339001.
- Tsunemi, T., T.D. Ashe, B.E. Morrison, K.R. Soriano, J. Au, R.A.V. Roque, E.R. Lazarowski, V.A. Damian, E. Masliah, and A.R. La Spada. 2012. PGC-1 α rescues Huntington's disease proteotoxicity by preventing oxidative stress and promoting TFEB function. *Science Translational Medicine.* 4:142ra97. doi:10.1126/scitranslmed.3003799.
- Vainshtein, A., L.D. Tryon, M. Pauly, and D.A. Hood. 2015. Role of PGC-1 α during acute exercise-induced autophagy and mitophagy in skeletal muscle. *Am J Physiol, Cell Physiol.* 308:C710–C719. doi:10.1016/j.cmet.2007.11.004.
- Zhu, W., G. Swaminathan, and E.D. Plowey. 2014. GA binding protein augments autophagy via transcriptional activation of BECN1-PIK3C3 complex genes. *autophagy.* 10:1622–1636. doi:10.4161/auto.29454.

Concern 5: “They also need to explain why mitochondria are dysfunctional when PGC1a is down-regulated - due to defective mitophagy or is it due to reduced biogenesis? How does the mitochondrial defect relate to the impact on EMT with loss of epithelial phenotype? Overall more work is needed to "drill down" on each of these findings to make the work sufficiently compelling for publication.”

Answer: Though PGC-1 α is known to promote mitochondrial biogenesis through NRF1-dependent induction of Tfam, our data suggests that mitochondrial biogenesis is not significantly affected by PGC-1 α loss of function in RPE. Indeed mitochondria, though abnormal, are still prominently observed in our in vivo and in vitro systems (Figures 1C and 6E) and gene expression analysis shows no repression in Polg and Tfam (Figure S1B) which point to defective mitophagy has the main cause of defective mitochondria. Evidence of deficient mitophagy include our report of impaired autophagic flux combined with hindered AMPK activation following PGC-1 α silencing (Williams et al., 2017; Laker et al., 2017) which are consistent with our observed accumulation of abnormal mitochondria (Figure 1C), increased mitochondrial superoxide production (Figure 1G) and defective mitochondrial function (Figure 1F). Definitive determination of mitophagic flux by use of specific tools such as MitoTimer or Mt-Keima is unfortunately not possible in our experimental system as our lentivirally-transduced cells constitutively express the fluorescent marker eGFP.

We agree with the reviewer that insights on the intersecting processes by which PGC-1 α , oxidative metabolism and EMT interact are of high interest although outside the scope of this manuscript. We are currently investigating the role of metabolic reprogramming in promoting

EMT in RPE. Once completed, this study should provide important information on the causal role of PGC-1s and RPE metabolic dysfunction in EMT.

References:

- Williams, J.A., K. Zhao, S. Jin, and W.-X. Ding. 2017. New methods for monitoring mitochondrial biogenesis and mitophagy in vitro and in vivo. *Experimental Biology and Medicine*. 242:781–787. doi:10.1083/jcb.201603039.
- Laker, R.C., J.C. Drake, R.J. Wilson, V.A. Lira, B.M. Lewellen, K.A. Ryall, C.C. Fisher, M. Zhang, J.J. Saucerman, L.J. Goodyear, M. Kundu, and Z. Yan. 2017. Ampk phosphorylation of Ulk1 is required for targeting of mitochondria to lysosomes in exercise- induced mitophagy. *Nat Rev Mol Cell Biol*. 19:121–135. doi:10.1038/s41467-017-00520-9.

Reviewer #2:

“Mariana Aparecida Brunini Rosales et al show that PGC-1 α depletion leads to impaired autophagy and epithelial-mesenchymal transition in RPE cells resembling age-related macular degeneration (AMD). This is excellent manuscript. I liked very much to read it. Hypothesis is very relevant. Study is very well planned. Data is strong and convincing. Manuscript is very well written. Discoveries may really open new ways to find better treatments to prevent or treat AMD. I have only minor comments to improve manuscript.”

We thank the reviewer for his/her interest in our work and support of its promise.

Concern 1: “In Fig. 6A-D indicate organelle or tissue changes by arrow.”

Answer: As suggested we have revised Fig 6 and added relevant annotations.

Concern 2: “Autophagosomes and autolysosomes were not calculated to confirm autophagy flux by TEM, but this can be replaced by citing and discussing the recent publication: Loss of NRF-2 and PGC-1 α genes leads to retinal pigment epithelium damage resembling dry age-related macular degeneration. Felszeghy S, Viiri J, Paterno JJ, Hyttinen JMT, Koskela A, Chen M, Leinonen H, Tanila H, Kivinen N, Koistinen A, Toropainen E, Amadio M, Smedowski A, Reinisalo M, Winiarczyk M, Mackiewicz J, Mutikainen M, Ruotsalainen AK, Kettunen M, Jokivarsi K, Sinha D, Kinnunen K, Petrovski G, Blasiak J, Bjørkøy G, Koskelainen A, Skottman H, Urtti A, Salminen A, Kannan R, Ferrington DA, Xu H, Levonen AL, Tavi P, Kauppinen A, Kaarniranta K. *Redox Biol*. 2019;20:1-12. doi: 10.1016/j.redox.2018.09.011.”

Answer: Because of the wide range of phenotypic anomalies we observed in the RPE of our experimental mice, we found quantification of autophagic organelles particularly challenging. As recommended by the reviewer, we are now discussing the close similarity of our findings with the one reported by Felszeghy S, et. al. (Lines 296-302). This recent publication is indeed quite complementary to our manuscript showing that RPE specific deletion of PGC-1 α promotes mitochondria/lysosomal dysfunction contributing to RPE dedifferentiation.

Reviewer #3:

“In the current manuscript, the authors convincingly show mitochondrial dysfunction and oxidative damage after repressing PGC-1 α expression. ... The study builds on thoroughly

generated, reproduced data. Data are presented in a comprehensive and clear way. References are provided in a complete and meaningful way. Workplan and the resulting data are conceivable and convincing. Each figure is designed such, that it complies with the experimental plan of the study. The results are well discussed in the context of the current level of understanding. Taken together, this is a convincing manuscript that merits publication.”

We thank the reviewer for his/her positive and detailed evaluation of our work.

Comment 1: “As human RPE cells as well mice are used, the coverage between mouse sequence and human sequence of PCG1a (line 495) in the overexpression experiment would be of interest.”

Answer: As indicated by phylogenetic analysis, PGC-1 α is highly conserved in vertebrates particularly in the functional domains (LeMoine et al., 2010). Previous studies from us and others have shown that murine and human PGC-1 α share very similar functions and are interchangeable (Diman et al., 2016, Iacovelli et al., 2016). We have now referencing the papers below in our methods section (Line 537).

References:

LeMoine, C.M.R., S.C. Loughheed, and C.D. Moyes. 2010. Modular evolution of PGC-1alpha in vertebrates. *J. Mol. Evol.*

Diman, A., J. Boros, F. Poulain, J. Rodriguez, M. Purnelle, H. Episkopou, L. Bertrand, M. Francaux, L. Deldicque, and A. Decottignies. 2016. Nuclear respiratory factor 1 and endurance exercise promote human telomere transcription. *Sci. Adv.*

Iacovelli, J., G.C. rowe, A. khadka, D. diaz-Aguilar, C. Spencer, Z.P. Arany, and M. Saint-Geniez. 2016. PGC-1alpha induces human RPE oxidative metabolism and antioxidant capacity. *Invest Ophthalmol Vis Sci.*

Comment 2: “EMT can be associated with reduced proliferation (i.e. some forms of cancer). I wonder whether cell numbers change after PCG1a silencing, I would have expected no differences in this case, because the energy metabolism switch and autophagy already require great effort for RPE cells. This is addressed in fig 3I: cell numbers increased. However, this is no further investigated or discussed.”

Answer: We have now addressed this comment by comparing the expression levels of the proliferation marker, PCNA, between shCtrl and shPGC-1 α cells at days 7, 14 and 21. We found that PCNA levels were higher in shPGC-1 α cells compared to shCtrl at day 7 but showed no difference at day 14 and 21. EMT in shPGC-1 α cells was most pronounced at day 21 and this coincides with our finding that PCNA levels were no longer increased in shPGC-1 α cells compared to shCtrl, thus suggesting that, as the shPGC-1 α cells become progressively more mesenchymal, there is also a reduction in proliferation (New Figure S2C).

Comment 3: “Fig 1E) CSA activity increases at day 14, then decreases at 21 days for shPGC1A. Please discuss!”

Answer: As discussed in our manuscript, we used CSA to measure mitochondrial mass however it appears that CSA does not correlate with mitochondrial content in metabolically defective RPE cells. We believe that changes in CSA represent a compensatory mechanism to sustain citrate production and/or support cell proliferation. As noted by the reviewer, there is a slight reduction in CSA activity in shPGC1A cells from day 14 to day 21. Interestingly our new evaluation of cellular proliferation does indicate high proliferative activity at day 7 in shPGC1A cells but not at days 14 and 21 (Figure S2C), which would support a role for citrate production for cell division (at the earliest time-point investigated). However we consider this interesting concept to be highly speculative at this stage. More definitive evidence on a role for citrate production in assisting RPE proliferative capacity would require significant investigations involving metabolomic profiling which is outside the scope of this manuscript. However, we did modify this point of discussion to integrate our new evaluation of cell proliferation (line 338).

Comment 4 “Figure 2B) It appears more suitable to adjust the order of the relative expression of genes within the graphs to the text as; LAMP1, MAP1LC3B, WIPI, ATG4D, ATG9B.”

Answer: The change has been made as suggested.

Comment 5: Figure 2E) Why are there two different western blotting bands for AAS and untreated conditions? To show two different repeats?

Answer: The reviewer is correct. As the results were quite unexpected we elected to show two sets of independent samples in the final figure and the quantification was performed on 4 independent samples. We are now complementing this figure with direct evaluation of autophagic flux further demonstrating impaired autophagy following PGC-1 α silencing (see response to Reviewer #1, Concern 1).

Comment 6: Figure 2H) The label for " Untreated versus AAS" similar to graphs Figure 2 E and Fig 2F is missing (Headline).

Answer: Figure 2H shows increased APOE expression in PGC-1 α silenced cells at day 14 of differentiation and cells were not amino acid starved. As explained in our results section (line 183): “As susceptibility to oxidative stress and impaired autophagy was noticed in primary RPE cells from AMD patients accompanied by upregulation of AMD-associated genes such as apolipoprotein E (APOE) expression (Golestaneh et al., 2017), we quantified the expression of APOE and found its protein level increased in shPGC-1 α RPE cells at 14 days (Fig. 2H).” To avoid confusion we have added a title to this figure.

Comment 7: Figure 3 E, F, G, H) -> For western blotting experiments GAPDH bands are not equal (lower in the treated panel). Did the authors check for equal protein concentration?

Answer: As described in our methods, all protein lysates were quantified using the bicinchoninic acid (BCA) method and the same protein content (30 μ g/sample) was loaded. We did notice also that GAPDH was reduced in shPGC1A cells at 21 days which could be explained by the drastic phenotypic change between control and experimental cells and potential alteration of standard housekeeping proteins. However, since our findings show increased

expression of EMT-proteins, normalizing to this protein would not affect the conclusions drawn that are further supported by our gene analysis (Fig 3D).

Comment 8: “Figure 5 D and F) "UV" should be changed to "μV" (ERG graph)”

Answer: We have corrected the figure.

Comment 9: “Figure 7 A) Mitochondrial/autophagic dysfunction and oxidative damage is shown in vitro. The authors show markers for oxidative phosphorylation and mesenchymal transition in vivo. How about any data to show autophagic dysfunction based on in vivo data?”

Answer: Monitoring of autophagic flux in vivo is extremely challenging and requires the use of reporter mice (Mizushima, 2009; Yoshii and Mizushima, 2017). Our preliminary evaluation of the autophagosome content in vivo does indicate that the number of LC3 puncta are reduced in the RPE of AAV-cre injected mice compared to controls. However, because of the regionality of the lesions and wide range of phenotypic changes observed (Figure 6), quantification was technically challenging. As suggested by Reviewer #2 we are now further discussing the similarity of our phenotype to the one described in NRF2/PGC-1α double knock-out animals by Felszeghy and colleagues (Felszeghy et al., 2019) (see comment to reviewer #2)

References:

- Mizushima, N. 2009. Methods for monitoring autophagy using GFP-LC3 transgenic mice. *Meth. Enzymol.* 452:13–23.
- Mizushima, N., T. Yoshimori, and B. Levine. 2010. Methods in Mammalian Autophagy Research. *Cell.* 140:313–326.
- Felszeghy, S., J. Viiri, J.J. Paterno, J.M.T. Hyttinen, A. Koskela, et al. 2019. Loss of NRF-2 and PGC-1α genes leads to retinal pigment epithelium damage resembling dry age-related macular degeneration. *Redox Biol.* 20:1–12.

Comments regarding the discussion:

“- Discussion on the age dependency of degenerating mouse retina is given for the AAV-cre mice. Discussion on the age dependency of results of RPE silenced for PGC-1α, however, is missing. Based on the in vitro data using 7, 14 and 21 days, this appears attractive.”

Answer: We agree with the reviewer that the results generated by our in vivo and in vitro experimental systems both agree with the concept that PGC-1α loss triggers a progressive degenerative process in RPE culminating in EMT. However we want to avoid excessive extrapolation from our longitudinal analysis, as our findings are correlative (see comments from reviewer #1, concern #2). We do mention in the discussion (Lines 290 and 356) that metabolic/autophagic dysfunction precede EMT, consistent with the emerging roles of oxidative stress, metabolic-epigenetic interplay and the ubiquitin-binding proteins like p62/SQSTM1 in EMT.

“- Functional data (ERG) could be discussed deeper in the discussion part? How do functional and histological data relate?”

Answer: We have now added a new paragraph in the discussion on the insights provided by the visual function analysis to our histological evaluations (lines 402-411).

“- With respect to Iba-1 as a marker revealing an immune response, is there a pattern?”

Answer: We have not investigated the potential activation and/or recruitment of immune cells in our animal model, though it is likely that the severe retinal degeneration we observe at 4 month would be associated with migration of microglia at the site of the lesions as generally observed in models of RPE dysfunction/atrophy (Yao et al., 2015; Kim et al., 2014, Felszeghy et al., 2019).

References:

Yao, J., L. Jia, N. Khan, C. Lin, S.K. Mitter, M.E. Boulton, J.L. Dunaief, D.J. Klionsky, J.-L. Guan, D.A. Thompson, and D.N. Zacks. 2015. Deletion of autophagy inducer RB1CC1 results in degeneration of the retinal pigment epithelium. *autophagy*. 11:939–953.

Kim, S.-Y., H.-J. Yang, Y.-S. Chang, J.-W. Kim, M. Brooks, E.Y. Chew, W.T. Wong, R.N. Fariss, R.A. Rachel, T. Cogliati, H. Qian, and A. Swaroop. 2014. Deletion of Aryl Hydrocarbon Receptor AHR in Mice Leads to Subretinal Accumulation of Microglia and RPE Atrophy. *Invest Ophthalmol Vis Sci*. 55:6031. doi:10.1167/iovs.14-15091.

Felszeghy, S., J. Viiri, J.J. Paterno, J.M.T. Hyttinen, A. Koskela, et al. 2019. Loss of NRF-2 and PGC-1 α genes leads to retinal pigment epithelium damage resembling dry age-related macular degeneration. *Redox Biol*. 20:1–12.

“- Does photoreceptor morphology change upon PGC-1a silencing?”

Answer: As shown in Figure 6, PGC-1 α silencing in RPE triggers a progressive and regional RPE dysfunction associated with a dramatic degeneration of the photoreceptors located above. Morphological changes are characterized by complete loss of outer-segment and severe thinning of the outer nuclear layer. We have annotated Figure 6 to further describe the changes observed.

“- PGC-1 α has a role in serving as a switch between mitochondrial biogenesis and oxidative damage by controlling the mitochondrial levels of ROS (Kaarniranta et al., 2018). What do the authors conclude with respect of a potential switch mechanism and effect on the mesenchymal transition?”

Answer: As a global regulator of mitochondrial health, PGC-1 α activate the expression of a core genetic program needed for mitochondrial DNA replication, mitochondrial dynamics, electron transport chain protein expression, mitochondrial OXPHOS and fatty acid oxidation. In addition to promoting mitochondrial function, PGC-1 α concurrently induces many cytoplasmic and mitochondrial antioxidant enzymes to counterweigh the endogenous production of reactive oxygen species due to increased mitochondrial function.

PGC-1 α therefore does not act as a switch per se as it promotes both mitochondrial function and antioxidant capacity. As discussed in our manuscript, we pose that impaired mitochondrial function and autophagy are the underlying causes of EMT in RPE by promoting the induction of EMT-transcription factors (Lines 341-380). We are currently further investigating the precise metabolic and molecular mechanisms of EMT induction by metabolic rewiring and we will report on our findings in the near future.

References:

- Iacovelli, J., G.C. rowe, A. khadka, D. diaz-Aguilar, C. Spencer, Z.P. Arany, and M. Saint-Geniez. 2016. PGC-1 α induces human RPE oxidative metabolism and antioxidant capacity. *Invest Ophthalmol Vis Sci*.
- Satish, S., H. Philipose, M.A.B. Rosales, and M. Saint-Geniez. 2018. Pharmaceutical Induction of PGC-1 α Promotes Retinal Pigment Epithelial Cell Metabolism and Protects against Oxidative Damage. *Oxidative Medicine and Cellular Longevity*. 2018

“- Line 278 The authors state: "Our longitudinal analysis in vitro indicates that PGC-1 α loss alters primarily RPE metabolic and autophagic functions followed by EMT induction." How do these patterns as a consequence of loss of PGC-1 α in retinal pigment epithelial (RPE) cells trigger mitochondrial/autophagic dysfunction and oxidative damage mechanistically? “

Answer: As mentioned above, PGC-1 α controls the activity of multiples transcription factors (PPAR, ERR α , NRFs, AP1, ...) and consequently regulates (directly or indirectly) the expression of a family of genes including those associated with aerobic respiration, redox function, response to oxidative stress. With regards to RPE, our group has previously shown that PGC-1 α is strongly induced during RPE maturation, promotes RPE metabolic function through induction of mitochondrial complex proteins and resistance to cytotoxic stress by inducing essential mitochondrial and cytoplasmic anti-oxidant enzymes (Iacovelli et al., 2016; Satish et al., 2018). Conversely, PGC-1 α loss, as demonstrated in our manuscript, represses critical antioxidant enzymes (Figure 1I) and promotes RPE metabolic dysfunction and accumulation of ROS. We have added this comment to the discussion (Line 309).

“- Line 285 Mitochondrial oxidative damage and/or decreased ATP levels can directly impair lysosomal activity (Demers-Lamarche et al., 2016) and defective lysosomal degradation can reciprocally cause the accumulation of defective mitochondria (Osellame et al., 2013) Did the authors check ATP level of the cells after silencing PGC-1 α ?”

Answer: We are now providing additional data (Fig S1E) showing reduced ATP content following PGC-1 α silencing. Interestingly while significant, the ATP loss is less robust at days 14 and 21 compared to day 7 which could be explained by the increased basal glycolytic activity (see Fig S1D) compensating for the mitochondrial dysfunction.

“- Line 295 The rise in total ROS appeared to precede the detection of higher mitochondrial superoxide production which could be explained by an increase in mitochondrial calcium influx following mitochondrial stress (Ahmad et al., 2013) or reflect enhanced mitochondrial production of hydrogen peroxide and hydroxyl radical, both detectable with the probe used. Did the authors check for mitochondrial calcium influx and/or hydrogen peroxide and hydroxyl radical or is it an assumption?”

Answer: We have not directly measured mitochondrial calcium influx and/or hydrogen radical and only proposed these hypotheses in the discussion. To avoid confusion we have edited the comment as follows: “The rise in total ROS appeared to precede the detection of higher mitochondrial superoxide production which could be explained by an increase in mitochondrial

calcium influx following mitochondrial stress (Ahmad et al., 2013) or reflect enhanced mitochondrial production of hydrogen peroxide and hydroxyl radical (Kirkland and Franklin, 2001).” (Lines 320-324)

“- Line 301 PGC-1 α is well known as a major transcriptional inducer of mitochondrial biogenesis. However, PGC-1 α silencing in RPE did not alter the expression of the mitochondrial replication-related genes TFAM and POLG expression or grossly reduced mitochondrial content. If mitochondrial protein composition is not severely affected, how can loss of mitochondrial dysfunction be mechanistically explained?”

Answer: As explained above (see response to Reviewer #1, Concern 5), our results strongly suggest that PGC-1 α silencing impairs mitochondria clearance thereby promoting the accumulation of damaged and dysfunctional mitochondria.

“- Line 324 ...autophagic flux and promoting the accumulation of defective organelles. "Importantly our investigation of the faulty autophagic step in PGC-1 α -deficient RPE cells point to the inability of AMPK to be activated upon starvation suggesting the existence of a positive feedback mechanism between AMPK and PGC-1 α regulating RPE metabolism and autophagic functions. Is there any reference supporting this conclusion?”

Answer: The current literature generally indicates that AMPK has an upstream regulator of PGC-1 α function and expression (Cantó and Auwerx, 2009) so our findings that AMPK is no longer activated by amino acid starvation in PGC-1 α silenced RPE cells is quite unexpected and novel. Previous investigations of the AMPK/PGC-1 α axis have used pharmacological activators of AMPK such as AICAR or Metformin in PGC-1 α deficient systems to delineate the functions of AMPK dependent on PGC-1 α , but to our knowledge this is the first report of altered AMPK response to amino acid withdrawal secondary to PGC-1 α loss of function.

References:

Cantó, C., and J. Auwerx. 2009. PGC-1 α , SIRT1 and AMPK, an energy sensing network that controls energy expenditure. *Current Opinion in Lipidology*. 20:98-105.

April 30, 2019

RE: Life Science Alliance Manuscript #LSA-2018-00212-TR

Dr. Magali Saint-Geniez
Schepens Eye Research Institute
Department of Ophthalmology, Harvard Medical School
20 Staniford St
Boston, MASSACHUSETTS 02114

Dear Dr. Saint-Geniez,

Thank you for submitting your revised manuscript entitled "Loss of PGC-1 α in RPE induces mesenchymal transition and promotes retinal degeneration". As you will see, while reviewer #1 regrets that no mitophagy assay was conducted, the reviewers are overall supportive of publication of your work. We would thus be happy to publish your paper in Life Science Alliance pending final revisions necessary to meet our formatting guidelines.

- please indicate in the legends which statistical test was used (you mention this in the methods section, but different tests have been used and it would thus be helpful to be specific in the legends)
- note that Fig3 panel H is missing in the legend, please add
- please mention arrows in the legend to figure 2
- the origin of higher magnification (inserts) is not evident in all figures, please check figure 2 and 6 (and please note that boxes for Fig 6A do not correspond well-enough to zoomed parts)

A. FINAL FILES:

B. MANUSCRIPT ORGANIZATION AND FORMATTING:

Sincerely,

Reviewer #1 (Comments to the Authors (Required)):

This is a re-review of a manuscript submitted previously which has since been revised to address the points I raised previously. The authors have robustly addressed most of the questions raised except concern #5 about actually measuring mitophagy. They claim that this is not possible since their system constitutively expresses GFP although GFP would not interfere with mt-Keima. The field is getting more rigorous about claiming to observe mitophagy based on metrics that are indirect measures of mitophagy such as those used here. However, this is perhaps an editorial decision given that other concerns have been addressed.

Reviewer #2 (Comments to the Authors (Required)):

Authors have answered fully to my suggestions. I have no any other demands. This is excellent work.

May 9, 2019

RE: Life Science Alliance Manuscript #LSA-2018-00212-TRR

Dr. Magali Saint-Geniez
Schepens Eye Research Institute of Massachusetts Eye and Ear
Department of Ophthalmology, Harvard Medical School
20 Staniford St
Boston, MASSACHUSETTS 02114

Dear Dr. Saint-Geniez,

Thank you for submitting your Research Article entitled "Loss of PGC-1 α in RPE induces mesenchymal transition and promotes retinal degeneration". It is a pleasure to let you know that your manuscript is now accepted for publication in Life Science Alliance. Congratulations on this interesting work.

DISTRIBUTION OF MATERIALS:

Again, congratulations on a very nice paper. I hope you found the review process to be constructive and are pleased with how the manuscript was handled editorially. We look forward to future exciting submissions from your lab.

Sincerely,
